# MUTUAL INFORMATION PRESERVING NEURAL NETWORK PRUNING

## ABSTRACT

Pruning has emerged as one of the primary approaches used to limit the resource requirements of large neural networks (NNs). Since the proposal of the lottery ticket hypothesis, researchers have focused either on pruning at initialization or after training. However, recent theoretical findings have shown that the sample efficiency of robust pruned models is proportional to the mutual information (MI) between the pruning masks and the model's training datasets, *whether at initialization or after training*. In this paper, we introduce Mutual Information Preserving Pruning (MIPP), a structured activation-based pruning technique applicable before or after training. The core principle of MIPP is to select nodes in a way that conserves MI shared between the activations of adjacent layers, and consequently between the data and masks. Approaching the pruning problem in this manner means we can prove that there exists a function that can map the pruned upstream layer's activations to the downstream layer's, implying re-trainability. We demonstrate that MIPP consistently outperforms baselines, regardless of whether pruning is performed before or after training.

## 1 INTRODUCTION

It is well-established that to limit a model's resource requirements while maintaining its performance, it is preferable to *prune* and re-train a large model of high accuracy rather than train a smaller model from scratch (LeCun et al., 1989; 1998; Li et al., 2017; Han et al., 2015). The lottery ticket hypothesis demonstrated that this was due to the existence of performant dense subnetworks embedded in overparameterized models at initialization Frankle and Carbin (2019). This discovery motivated a new body of research on *pruning at initialization (PaI)*, such as SNIP (Lee et al., 2019), GraSP (Wang et al., 2022), SynFlow (Tanaka et al., 2020), and ProsPr (Alizadeh et al., 2022) to name a few. Subnetworks identified using these methods perform worse than those obtained through

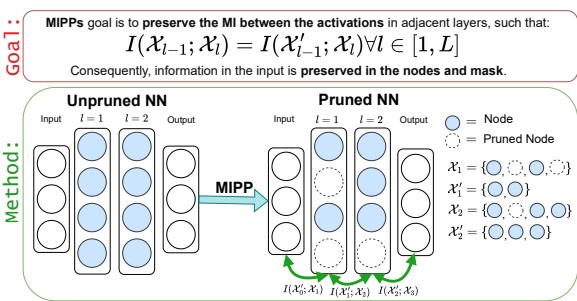

Figure 1: We introduce MIPP via an illustration. MIPP is a pruning method that acts to preserve the mutual information (MI) between the activations in adjacent layers. In turn, this leads to a pruned network representation whose nodes and mask effectively capture the information contained in the data.

*pruning after training (PaT)*, even when using straightforward approaches like iterative magnitude pruning (IMP) Frankle et al. (2021). Kumar et al. (2024)'s PAC-learnability result provided an information-theoretic justification for this, demonstrating that the sample efficiency of a pruned learning algorithm is proportional to the effective parameter count, which can be calculated by summing the number of unmasked parameters and the mutual information (MI) shared between the pruning mask and training data. Kumar et al. (2024) argues that to maximize the MI term, it is essential that training occurs, leading to the poor performance of the state-of-the-art (SOTA) PaI methods.

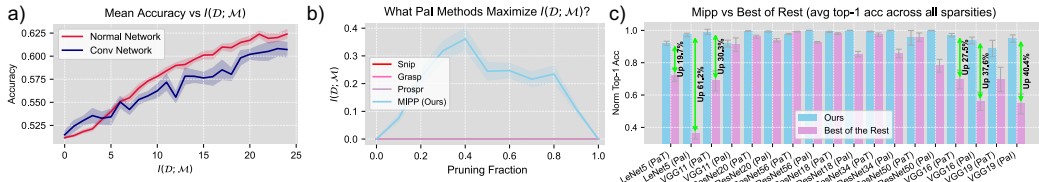

Figure 2: *a)* Graphical representation of how MI between the mask and the data affects the test accuracy of a small convolution-based and standard NN: we observe that by maximizing MI, the classification accuracy increases. The experiments are based on synthetic data; for full details refer to Appendix E.1. *b)* A study examining how pruning masks, created using various PaI methods and applied to a small synthetic network, affect the values of $I(\mathcal{D}; \mathcal{M})$. For full details about these experiments, please refer to Appendix E.2. *c)* Comparison of MIPP's accuracy averaged across all sparsity ratios to the best-performing baseline for each model-dataset combination; "best of rest" is also averaged across all sparsity ratios. MIPP outperforms the best of the rest significantly, as at high sparsities, they are all much more prone to layer collapse. PaT baselines: OTO, IMP, SOSP-H, ThiNet. PaI baselines: IterSNIP, IterGrasP, ProsPr, SynFlow. Results show the mean across 5 runs; error bars indicate 95% confidence intervals.

To provide an intuition for this PAC-learnability result by Kumar et al. (2024), Figure 2.a illustrates the improved accuracy that results from maximizing MI between the pruning mask and the training data in both a standard neural network (NN) and a convolutional variant. For these experiments, the data were synthetic and masks derived before training, so MI values were obtained analytically. Given the result presented in Kumar et al. (2024), supported in part by Figure 2.a, we conclude that maximizing MI shared between the pruning mask and training data is a sensible objective when pruning.

As originally argued by Kumar et al. (2024), optimizing this objective is expected to restrict us to a PaT approach; without training, we have no reason to expect that the weights or pruning masks will exhibit any correlation with the data. While this certainly holds for data-independent pruning schemes, such as magnitude PaT or other PaI solutions (like those presented by Tanaka et al. (2020); Patil and Dovrolis (2021); Pham et al. (2024)), it may not be universally true. For example, consider taking an activation-based approach. The activations at each layer of a NN are a function of the activations preceding them, or of the input data. If the NN is sufficiently expressive, these activations should contain all the information in the data, whether training has occurred or not. Therefore, if we can define a mask that preserves all the information in the activations, it should transfer to the data and maximize our objective, even at initialization.

Consequently, we introduce Mutual Information Preserving Pruning (MIPP), a structured activation-based pruning technique applicable before or after training. MIPP ensures that MI shared between activations in adjacent layers is preserved during pruning (please refer to Figure 1[1].). Rather than ranking nodes and selecting the top-$k$, MIPP uses the transfer entropy redundancy criterion (TERC) (Westphal et al., 2024) to dynamically prune nodes whose activations do not transfer entropy to the downstream layer. We will show that pruning in this manner ensures the existence of a function that can re-construct the downstream layer from the pruned upstream layer. Moreover, we will demonstrate that MIPP establishes pruning masks whose MI with the training data has a maximal upper bound. This is because MIPP dynamically evaluates and removes redundant nodes in a manner dependent on those currently maintained in the network representation, a feature that is unachievable using static ranking-based pruning methods. To illustrate this visually, in Figure 2.b we show that only MIPP can derive pruning masks that have non-zero mutual information with the data for a synthetic NN characterized by nodes sharing redundant information. Finally, we demonstrate MIPP's utility beyond theoretical justification by presenting improved pruning results in both post- and pre-training domains, as shown in Figure 2.c. To summarize, the contributions of this work are as follows:

- We develop MIPP, a structured activation-based pruning method that preserves MI between the activations of adjacent layers in a deep NN.

---

[1]Saxe et al. (2019) demonstrated that the MI relating inputs and activations becomes infinite, for a more detailed discussion of this and its relation to our work, refer to Appendix H

- We prove that perfect MI preservation ensures the existence of a function, discoverable by gradient descent, which can approximate the activations of the downstream layer from the activations of the preceding pruned layer. Consequently, MIPP implies re-trainability.
- We prove that pruning using MIPP leads to a maximum upper bound on MI between the data distribution and the mask distribution (as defined in Kumar et al. (2024)).
- Through comprehensive experimental evaluation[2], we demonstrate that MIPP can effectively prune networks, whether they are trained or not.

## 2 RELATED WORK

**Pruning after training.** Traditional structured pruning methods employ metrics such as weight magnitude (Han et al., 2015; Li et al., 2017; Wang and Fu, 2023), weight gradient (LeCun et al., 1998; Molchanov et al., 2017), Hessian matrices (Hassibi and Stork, 1992; Peng et al., 2019; Wang et al., 2019; Nonnenmacher et al., 2022), and combinations thereof, to rank and then remove nodes up to a defined pruning ratio (PR). Although these methods were originally designed to be applied at the level of individual weights, they can be adapted for structured cases through non-lossy functions, such as L1-normalization Wang and Fu (2023). This can be carried out in either a global or local manner, the former involves ranking all the nodes in a network (Liu et al., 2017; Wang et al., 2019), while the latter is only applied to individual layers (Zhao et al., 2019; Sung et al., 2024). Global methods have been effective in determining layer-wise pruning ratios (Blalock et al., 2020). However, at high PRs, they experience layer-collapse, an undesirable final result in which an entire layer is pruned and an untrainable network is produced Tanaka et al. (2020). Traditional methods, including magnitude, gradient, and Hessian-based approaches, continue to represent the SOTA due to recent methodological refinements. Modern variations of such techniques are *iterative*, meaning that the model is trained, some fraction of the weights - lower than the final PR - are removed according to the methods described, and then the model is retrained and the process is repeated until the PR is reached Frankle and Carbin (2019); You et al. (2020). These methods are known to lead to highly performant models, while also being resistant to layer collapse. However, they are computationally expensive because they require multiple retraining sessions. In response, methods such as SOSP-H have been proposed Nonnenmacher et al. (2022). SOSP-H ranks and removes nodes in a traditional way, except for the fact that the metric employed is the Hessian Hassibi and Stork (1992). The Hessian is recognized as the most computationally expensive yet best-performing metric Molchanov et al. (2019). By employing a second-order approximation, its benefits can be leveraged in a computationally efficient manner. While MIPP acts globally, aligning with the methods discussed thus far, it is also activation-based, diverging from these competing techniques. ThiNet Luo et al. (2017) most closely resembles MIPP in terms of methodology, although it is known that it is unable to establish layer-wise PRs. Other information-theoretic pruning methods applied post-training include the variational information bottleneck approach of Dai et al. (2018), which prunes neurons by minimizing information flow; the rate–distortion perspective of Isik et al. (2022), which provides theoretical justification for pruning decisions; MINT by Ganesh et al. (2020), which leverages conditional mutual information between adjacent layers; and the layer-wise mutual-information–based transformer pruning method proposed by Fan et al. (2021).

**Pruning at initialization.** In contrast to pruning after training, pruning at initialization aims to identify and remove redundant parameters before the training process begins, thereby reducing computational overhead from the outset. Early approaches, such as SNIP Lee et al. (2019) and GraSP Wang et al. (2022), leverage sensitivity metrics based on gradients to determine which weights can be safely pruned. Nevertheless, when applied globally, such methods suffer from layer collapse. In response, Tanaka et al. (2020) developed an iterative method of PaI, which mirrors that described in the previous paragraph but without re-training. This reduced layer-collapse occurrence, and improved the performance when PaI. However, recent results have suggested that the performance of such methods is not due to the selected nodes, but rather the per-layer PRs. As demonstrated by Frankle et al. (2021) and Su et al. (2021), the performance of models established using SNIP, GraSP, and SynFlow is robust to the weight shuffling within layers. Nevertheless, this phenomenon was not repeated at ultra-high sparsity. Pham et al. (2024) argued that this was evidence that, when one aims to PaI, their objective should be to preserve the number of effective paths, as achieved in PHEW

---

[2]The code is available at the following URL: *[the entire codebase of MIPP will be made available upon publication]*.

and NpB Patil and Dovrolis (2021); Pham et al. (2024). These methods outperformed SNIP despite being data-independent. Nevertheless, they failed to attain results comparable to PaT, unlike ProsPr Alizadeh et al. (2022).

## 3 MUTUAL INFORMATION PRESERVING PRUNING AT A GLANCE

In this section, we introduce the required notation before a formal definition of MIPP. The function describing a layer $l$ in an NN can be written as follows: $f_l(\boldsymbol{x}_l^n) = \boldsymbol{x}_{l+1}^m = a(\boldsymbol{W}_l^{m \times n} \boldsymbol{x}_l^n + \boldsymbol{b}_l^m)$. In the above, a is an activation function, $\boldsymbol{W}_l^{m \times n}$ is a weight matrix, $\boldsymbol{b}_l^m$ the bias, and $\boldsymbol{x}_l^n$ is the input to that layer (LeCun et al., 1998; Goodfellow et al., 2016).

Structured pruning is the process of discovering per-layer binary vector masks ($\boldsymbol{m}_l^n$) that zero out weight matrix elements corresponding to a node or filter index. We will denote a pruned layer with a prime symbol (') (Fahlman and Lebiere, 1990). The set of all masks associated with a network is given by $\mathcal{M}_0$, while the function associated with a pruned layer can be written as: $f_l'(\boldsymbol{x}_l^n) = \boldsymbol{x}_{l+1}'^m = a(\boldsymbol{W}_l^{m \times n} \boldsymbol{x}_l^n \boldsymbol{m}_l^n + \boldsymbol{b}_l^m)$. By randomly sampling from the space of possible inputs and applying the function described by the NN, we form not only the inputs as random variables (RVs), but also all subsequent activations. We define $X_l^i$ as the RV associated with the activations of node $i$ in layer $l$. Meanwhile, the set $\mathcal{X}_l = \{X_l^0, X_l^1 \ldots X_l^n\}$ contains a RV for all of the $N$ neurons in layer $l$. We use $\mathcal{X}_0$ to indicate the input. If a pruning mask is multiplied with the weights, the activations associated with pruned nodes are set to zero, which can otherwise be seen as information theoretically null. We denote the set associated with a pruned layer as $\mathcal{X}_l'$. If multiple pruning runs are performed with different datasets, multiple pruning masks will be created. In this case, both our pruning mask and our data distribution can be viewed as RVs, $\mathcal{M}_0$ becoming $\mathcal{M}$, while $\mathcal{X}_0$ becomes $\mathcal{D}$. For a full table of notation please refer to Appendix A.

MIPP is founded on the idea that maximizing MI between realizations of the pruning mask and the data distribution, denoted as $I(\mathcal{D}; \mathcal{M})$, ensures effective pruning with minimal performance loss. To achieve this, MIPP preserves MI between adjacent layers throughout a network. More specifically, we aim to isolate masks $\boldsymbol{m}_l^n$, which combine with the weights to produce updated layers with some of the activations equal to zero. These null activations should not cause a reduction in the MI between the activations of adjacent layers. Our goal is to preserve the joint mutual information between the activations of adjacent layers before and after pruning, i.e., $I(X_l; X_{l+1}')$. In other words, the MI between masked adjacent layers after pruning should equal that of unmasked adjacent layers before pruning, quantifying how much of the downstream layer's representational content is retained after pruning the upstream layer. More formally, this can be expressed as follows: $\mathcal{M}_0 = \{\boldsymbol{m}_l^n \forall l \in [1, L] : I(\mathcal{X}_{l-1}'; \mathcal{X}_l) = I(\mathcal{X}_{l-1}; \mathcal{X}_l)\}$. We note that this equality is not assumed to hold exactly in practice. Instead, we use MI thresholds and rely on the empirical observation that removing individual units often does not significantly change MI, especially in redundant layers. The theoretical equality serves as a guiding principle rather than an operational requirement.

## 4 THEORETICAL MOTIVATION

We now motivate MIPP theoretically. As stated, we aim to design a method that preserves MI between activations such that $I(\mathcal{X}_{l-1}'; \mathcal{X}_l) = I(\mathcal{X}_{l-1}; \mathcal{X}_l)$. In this section, we point out two advantages of doing this. Pruning in this manner not only ensures re-trainability, but it also leads to an optimal upper bound on the value of MI between the data-distribution and the masks $I(\mathcal{D}; \mathcal{M})$ (as defined in Kumar et al. (2024)).

**Re-trainability.** We consider one-shot pruning with (re)-training: the objective remove nodes such that, after retraining, the pruned NN will achieve the same performance as the original. We argue that one way to achieve this would be to select a subset of nodes from each layer so that there exists a function, which, when applied to this subset, can still reconstruct the activations of the subsequent layer. We will then prove that the existence of this function preserves MI between the activations of these layers.

To illustrate this, we guide the reader through the following example. Consider the case in which we generate the expected outputs of our NN from the activations of the last layer. More formally, we write $\mathcal{X}_L = f_{L-1}(\mathcal{X}_{L-1})$. We now wish to prune the activations preceding the outputs. This entails

minimizing the number of nodes or the cardinality of the set $\mathcal{X}'_{L-1}$ in such a manner that there exists a function that can reliably re-form $\mathcal{X}_L$. Furthermore, this function should be discoverable by gradient ascent. More formally, we would like to derive $\mathcal{X}'_{L-1}$ such that $\mathcal{X}_L = \sup_{g \in \mathcal{F}} g(\mathcal{X}'_{L-1})$, where $\mathcal{F}$ denotes the set of all candidate functions under consideration (e.g., the function class representable by the network layer). While this formulation reveals little in the way of a potential pruning operation, using the following theorem, we relate it to the MI-based objective: $I(\mathcal{X}'_{l-1}; \mathcal{X}_l) = I(\mathcal{X}_{l-1}; \mathcal{X}_l)$.

Consequently, in this work we aim to select a set of masks ($\mathcal{M}_0$) that increase sparsity while preserving MI between layers. This ensures that, for each pruned layer, there exists a function, discoverable by gradient descent, that effectively reconstructs the activations of the subsequent layer using those of the pruned layer. In other words, MIPP implies re-trainability under certain theoretical assumptions, which may not always hold in practice. We note that other methods demonstrate their ability to preserve retrainability through empirical results only Wang and Fu (2023).

**Maximizing $I(\mathcal{D}; \mathcal{M})$.** As discussed in the introduction, a sensible pruning objective is to maximize $I(\mathcal{D}; \mathcal{M})$. Westphal et al. (2024) proved that TERC, the method we use to preserve MI between layers, does so via the derivation of a bijective function. This implies that the activations of the upstream pruned layer $\mathcal{X}'_{l-1}$ can be used to produce the downstream layer $\mathcal{X}_l$ and vice versa. In this section, we present theoretical results showing that the existence of such a bijective function allows the derivation of a maximum upper bound on the achievable MI between the masks and datasets $I(\mathcal{D}; \mathcal{M})$.

**Theorem 1:** *If a pruning method preserves MI between layers activations then the upper bound on $I(\mathcal{D}; \mathcal{M})$ reaches its maximum. More formally: $I(\mathcal{X}'_{l-1}; \mathcal{X}_l) = I(\mathcal{X}_{l-1}; \mathcal{X}_l) \Leftrightarrow I(\mathcal{D}; \mathcal{M}) \leq H(\mathcal{D})$.*

*Proof.* See Appendix C.

As a result, when using MIPP there is a greater upper-bound on the value of $I(\mathcal{D}; \mathcal{M})$, which has been shown to be related to the models' accuracy and sample efficiency. However, how this quantity explicitly relates to generalization is more nuanced and is discussed in Appendix I.

# 5 MIPP

## 5.1 PRELIMINARIES

### 5.1.1 TRANSFER ENTROPY REDUNDANCY CRITERION WITH MI ORDERING

Before describing the method, we now provide a summary of TERC and its application to pruning, through the incorporation of an additional step for MI-based ordering.

**Node Pruning using TERC.** MIPP uses the transfer entropy redundancy criterion (TERC) (Westphal et al., 2024) to dynamically prune nodes whose activations do not transfer entropy to the downstream layer. As discussed in Section 3, we aim to preserve MI between the layers in our network. The problem of MI preservation is one well-studied in the feature selection community (Battiti, 1994; Peng et al., 2005; Gao et al., 2016). We chose TERC, as not only does it preserve MI with the target via a bijective function, but its temporal complexity is also linear in time with respect to the number of features (Westphal et al., 2024), a key property when working in highly dimensional feature spaces. In our case, rather than selecting features to describe a target, we are selecting nodes that transfer entropy to the following layer. Within this context, TERC can be summarized as follows: to begin, all nodes in the layer are assumed to be useful (and added to the non-pruned set). We then sequentially evaluate whether the reduction in uncertainty of the subsequent layer's activations is greater when a specific node is included in the unpruned set rather than excluded. More formally, for a node $X_l^i$ to be added the set of pruned nodes, it must satisfy the following condition $I(\mathcal{X}_{l-1}; \mathcal{X}_l) = I(\mathcal{X}_{l-1} \backslash X_l^i; \mathcal{X}_l)$. Otherwise, it is maintained in the network structure. This process is sequentially repeated for all nodes in the layer. As shown in Westphal et al. (2024), this simple technique will preserve MI between layers.

**MI Ordering.** Before applying TERC, we sort the nodes in the pruning layer in descending order of MI with the target (see Algorithm 2 in Appendix B). This step is motivated by Theorem 3 in Westphal et al. (2024). In particular, they prove that TERC alone selects unnecessary variables if there exists perfectly redundant variable subsets of different cardinalities. Ordering partly addresses this problem.

### 5.1.2 MUTUAL INFORMATION ESTIMATION

Unless restricting oneself to scenarios inapplicable to real-world data (e.g. discrete RVs), verifying the condition in Section 5.1.1 is computationally intractable. Consequently, we must approximate the condition using MI estimates, for which many methods have been developed (Moon et al., 1995; Paninski, 2003; Belghazi et al., 2018; van den Oord et al., 2019; Poole et al., 2019).

For the purposes of pruning, our MI estimates need to only be considered for comparison. Rather than using a method that is able to provide highly accurate estimates slowly (Franzese et al., 2024), we require one that emphasizes speed and consistent results. For these reasons, we adopt the technique presented in Covert et al. (2020), in which the authors demonstrate that MI between two random processes ($X$ and $Y$) can be approximated as the reduction in error estimation caused by using $X$ to predict $Y$. For further details on conditions under which this estimator approaches MI, please refer to Covert et al. (2020). More formally: $I(X; Y) \approx \mathbb{E}[l(f_\emptyset(\emptyset), Y)] - \mathbb{E}[l(f_X(X), Y)]$, where each f is some function approximated via loss l. If the target is discrete and degenerate, and a cross entropy loss is used, then this value is exactly equal to the ground truth MI (Gadgil et al., 2024). Even if the variables are continuous and a mean squared error loss is used, the above value approaches MI under certain circumstances.

### 5.2 PRESERVING THE MUTUAL INFORMATION BETWEEN ADJACENT LAYERS IN PRACTICE

In this section, we discuss how to use TERC to preserve MI between a pair of adjacent layers. As discussed, TERC with MI ordering dictates that, to remove a node, the following should be satisfied: $I(\mathcal{X}_{L-1} \backslash X_{L-1}^i; \mathcal{X}_L) = I(\mathcal{X}_{L-1}; \mathcal{X}_L)$. In Section 5.1.2, we described the method we used to estimate MI. By combining these representations, we can update the condition we wish to approximate as follows:

$$I(\mathcal{X}_{l-1}; \mathcal{X}_l) = I(\mathcal{X}_{l-1} \backslash X_{l-1}^i; \mathcal{X}_l) \quad \text{(original condition as in TERC)},$$
$$\mathbb{E}[l(f_{l-1}(\mathcal{X}_{l-1}), \mathcal{X}_l)] \geq \mathbb{E}[l(h_{l-1}(\mathcal{X}_{l-1} \backslash X_{l-1}^i), \mathcal{X}_l)] \quad \text{(updated condition)}. \tag{1}$$

Equation 1 represents the simplification possible when $I(X; Y) \approx \mathbb{E}[l(f(\emptyset), Y)] - \mathbb{E}[l(f(X), Y)]$ is substituted into $I(\mathcal{X}_{l-1}; \mathcal{X}_l) = I(\mathcal{X}_{l-1} \backslash X_{l-1}^i; \mathcal{X}_l)$. Our condition characterizes the case where node $X_{l-1}^i$ transfers no entropy to the following layer. The monotonicity of MI enforces that we have the equality seen in line one of Equation 1. When approximating this condition, as shown in the second line of Equation 1, we can no longer guarantee monotonicity. Therefore, we relax the equality to the inequality as indicated. Overall, our condition becomes a simple comparison of two losses quantifying two functions' ability to reconstruct the downstream layer. The definition of MI as presented in Covert et al. (2020) is applicable for any $f_{l-1}$ or $h_{l-1}$ discovered using function approximation. However, we need not fit a new function as we already posses $f_{l-1}$ exactly in the form of layer $l$ in our network. Here, $h_{l-1}$ denotes the function obtained by retraining layer $l-1$ to predict the downstream layer's activations with node $X_{l-1}^i$ masked in its input. If $\mathbb{E}[l(h_{l-1}(\mathcal{X}_{l-1} \backslash X_{l-1}^i), \mathcal{X}_l)]$ is equal to or drops below $\mathbb{E}[l(f_{l-1}(\mathcal{X}_{l-1}), \mathcal{X}_l)]$ our condition is satisfied and we can remove the node $X_{l-1}^i$.

Given the above condition, we now describe TERC with MI ordering: initially, we order the nodes in descending order of the loss achieved when using just this variable as input to predict the downstream layer $f_l(X_l^i)$. We now sequentially traverse the nodes in this order, similarly to Gadgil et al. (2024), masking them and re-training our layer (to find $h_l(\mathcal{X}_l \backslash X_l^i)$) to determine whether the loss function drops back below its original value $\mathbb{E}[l(f_{l-1}(\mathcal{X}_{l-1}), \mathcal{X}_l)] \geq \mathbb{E}[l(h_{l-1}(\mathcal{X}_l \backslash X_{l-1}^i), \mathcal{X}_l)]$. If it fails to recover, this implies that, without the activations of this node, we are unable to reconstruct the activations of the downstream layer. In this case, the variable is considered informative and should be retained in the network and in the set $\mathcal{X}'_{l-1}$. Otherwise, the node is removed. Once a node has been evaluated, the layer can be updated with the new trained function ($h_{l-1}$).

We have explained how MIPP is a structured pruning method that retains nodes whose activations transfer entropy to the next layer. The number of nodes maintained in the network is therefore dynamically dependent on those already selected, making us unable to set a pruning ratio in the traditional sense Hassibi and Stork (1992); Nonnenmacher et al. (2022). However, we wish to study MIPP at different degrees of sparsity. Consequently, we now briefly explain how we affect the pruning ratio discovered using MIPP. From the condition above, it is clear that for a node to be removed from the network, the loss must fall below the level achieved using all the activations, $\mathbb{E}[l(f_{l-1}(\mathcal{X}_{l-1}), \mathcal{X}_l)]$. To adjust the pruning ratio, we update this threshold by allowing it to take values that are regularly

spaced within the range $[\mathbb{E}[l(f_{l-1}(\mathcal{X}_{l-1}), \mathcal{X}_l)], \mathbb{E}[l(f_{l-1}(\emptyset), \mathcal{X}_l)]]$. If we are close to $\mathbb{E}[l(f_{l-1}(\emptyset), \mathcal{X}_l)]$, the condition for removing nodes is easily satisfied, and the sparsity ratio is high.

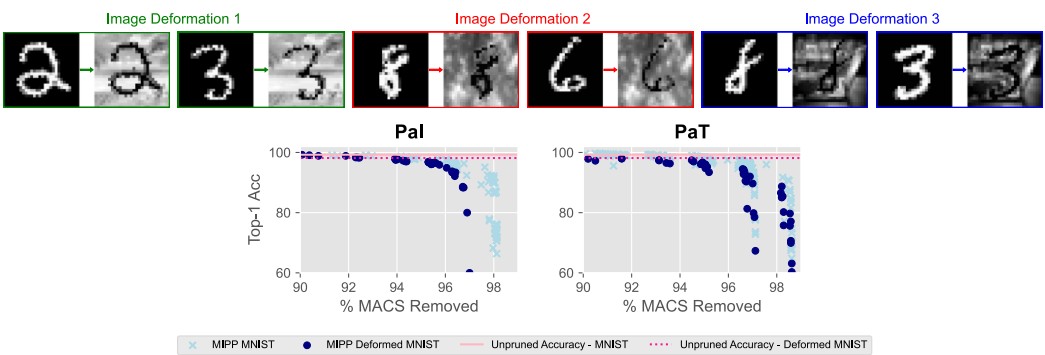

Figure 3: *Top.* Deforming MNIST for increased image complexity. These transformations were applied randomly with equal probability and then kept consistent during training, pruning, and re-training. *Bottom.* Changes in pruning ability of MIPP caused by image deformation. Results show test accuracy. Multiple deformation types are included to ensure robustness to the randomness arising from any single deformation, rather than to investigate varying deformation strengths.

### 5.3 Preserving the Mutual Information from Outputs to Inputs

Until now, we have focused on the use of TERC with MI ordering to preserve MI between the activations of adjacent layers. This process is repeated for each pair of layers. To prune the entire model, by preserving MI between pairs of layers, one could start from the input layer and move to the output layer or vice versa. In this section, like Luo et al. (2017), we argue for the latter option.

In an idealized theoretical construction, each layer in a neural network is an injective function of its predecessor, these pairs share perfect mutual information (MI), where $I(\mathcal{X}_{l-1}; \mathcal{X}_l) = H(\mathcal{X}_l)$, meaning network layers' activations can only reduce in entropy from inputs to outputs. In fact, when pruning from inputs to outputs, the first layer ($\mathcal{X}_1$) is pruned to reconstruct the second layer ($\mathcal{X}_2$), but since the second layer has not been pruned yet, it may retain irrelevant information that gets maintained in the first layer during pruning. Conversely, pruning from outputs to inputs begins with layer $\mathcal{X}_{L-1}$, where the pruned version $\mathcal{X}'_{L-1}$ preserves only information needed to reconstruct outputs. When pruning subsequent layers backwards, each layer only retains entropy required for the already-reduced next layer, ultimately ensuring the first layer retains only output-relevant information. More practically, this backward approach evaluates $I(\mathcal{X}'_{l-1}; \mathcal{X}'_l) = I(\mathcal{X}_{l-1}; \mathcal{X}'_l)$ rather than $I(\mathcal{X}'_{l-1}; \mathcal{X}_l) = I(\mathcal{X}_{l-1}; \mathcal{X}_l)$, which is a function of pruned layers' activations, mitigating the curse of dimensionality. Algorithm 1 formally describes MIPP's steps. In Appendix F.1, we also explain how MIPP can be used for feature selection.

## 6 Evaluation

In this section, we discuss the evaluation of MIPP, starting with the experimental settings and the datasets used. We selected MNIST, CIFAR-10, CIFAR-100, and TinyImageNet for their benchmark status enabling a comprehensive evaluation of MIPP while ensuring comparability across prior work (LeCun et al., 1989; Krizhevsky, 2009; University, 2015).

### 6.1 Models, datasets and baselines

We begin by applying our method to the simple LeNet5 architecture detecting variations of the MNIST dataset (LeCun et al., 1998). We then assess its ability to prune ResNet20, ResNet56, ResNet18, ResNet34, and VGG11 on the CIFAR10 dataset (He et al., 2016; Simonyan and Zisserman, 2015). Before then investigating ResNet50, VGG16, and VGG19 models networks trained on CIFAR100 (Krizhevsky, 2009). Finally, we also investigate a ResNet20 trained on TinyImageNet (University,

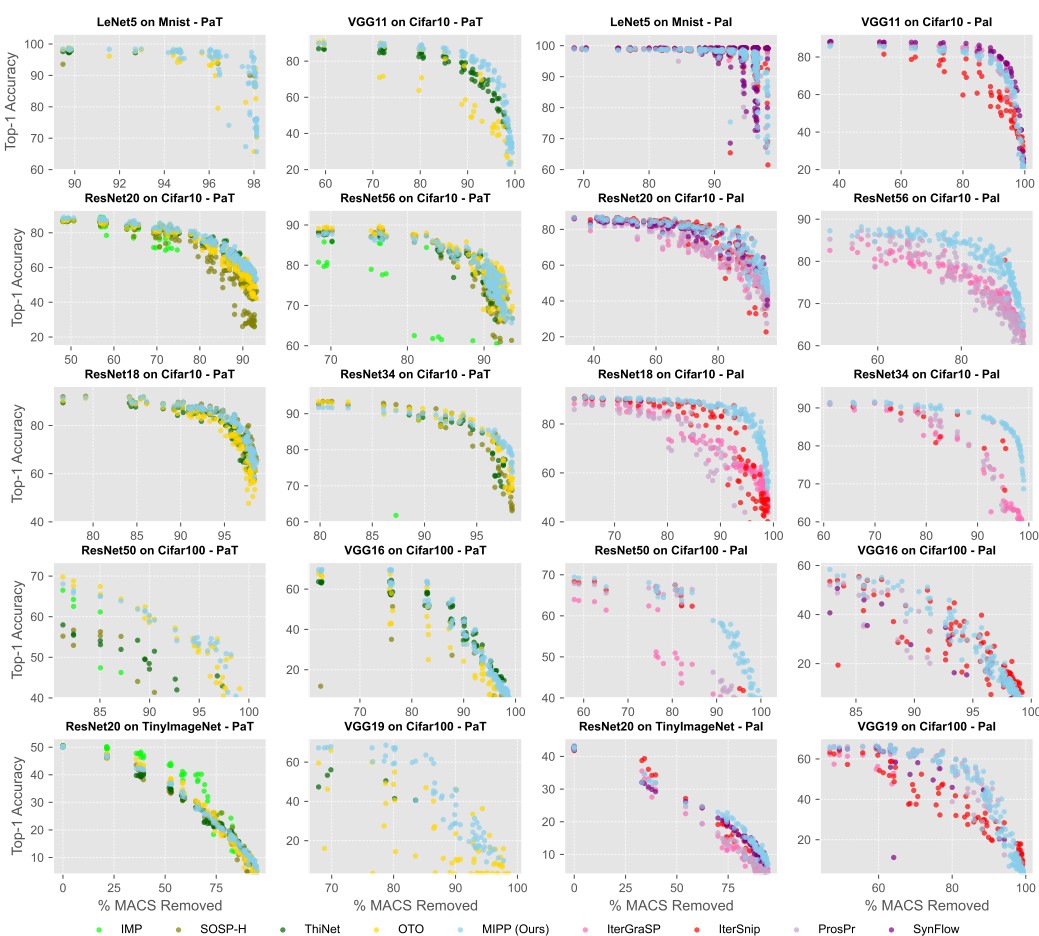

Figure 4: Comaprison of MIPP's ability to prune versus baselines both at initialization and after training. For clarity, we set an accuracy range to avoid viewing data points in which layer collapse has occurred.

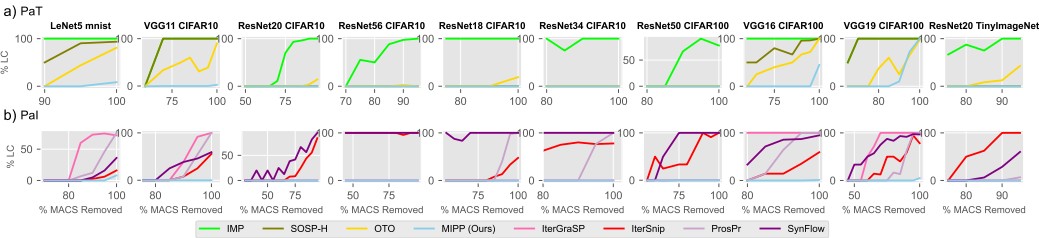

Figure 5: The percentage of runs that led to untrainable layer collapse. Specifically, we bin runs by the percentage of neurons removed, where one bin contains all the runs within a 5% increment. We then calculate the percentage of these runs that lead to layer collapse.

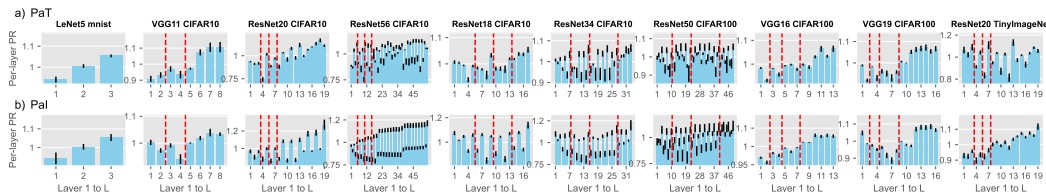

Figure 6: These experiments demonstrate the per-layer PR selected by MIPP. For the different layer-wise PRs we divide them by the average of all the layers in order to normalize. Red dotted lines indicate block end.

2015). When using MIPP to PaI, we compare to SynFlow (Tanaka et al., 2020), IterGraSP (Wang et al., 2022), IterSNIP (all 100 iterations) and ProsPr. Meanwhile, when using MIPP to PaT we compare to IMP Frankle and Carbin (2019), OTO You et al. (2020), ThiNet (Luo et al., 2017) and SOSP-H (Nonnenmacher et al., 2022). GraSP, SynFlow and SNIP are unstructured; to make them structured, we apply L1-normalization to all the weights associated with a node. MIPP selects nodes based on whether their activations transfer entropy to those of the subsequent layer. This approach inherently establishes a unique PR for each run, which we adopt as the global PR for our baseline methods. ThiNet cannot determine layer-wise PR; so we apply a uniform PR across all layers.

## 6.2 LeNet5 on varieties of MNIST

In this section, we begin by analyzing how increasing the complexity of the MNIST dataset impacts MIPP's performance. We then evaluate how MIPP compares to baseline methods when applied to a LeNet-5 architecture on MNIST.

Empirical evidence indicates that the utility of PaI may be limited to simple datasets (Frankle and Carbin, 2019; Frankle et al., 2021). We provide an information-theoretic argument with empirical evidence to explain this phenomenon. MIPP, and other effective PaI schemes, preserve and compress the information encoded in network activations. In untrained networks, these activations reflect the *entirety* of the information present in the input data. If these inputs are characterized by information relevant to the classification task, MIPP (and PaI more generally) remains applicable. For instance, in the MNIST dataset, the informative pixels assist the classification task, while the remaining pixels, on the outskirts of the image, are constantly black and contain no information. In such cases, our method selectively preserves the neurons whose activations correspond to informative pixels. On the other hand, the converse is also true; our method is inapplicable to models whose input data contains information not relevant for the classification task. Consequently, if the input data is complex, MIPP will preserve highly entropic activations over those that are useful for the downstream task, which can impede MIPP's PaI capabilities. To demonstrate this effect, in Figure 3 we present experiments that investigate the effects of deforming MNIST. We deliberately distort MNIST images, preserving the identifiability of the original digits (Figure 3) while making the formerly black pixels more entropic. In alignment with our hypothesis, we observe a reduction in our ability to prune an untrained network but not a trained network when the dataset complexity is increased.

Figure 4 demonstrates that MIPP performs at least as well as the baselines, regardless of whether PaI or PaT. Additionally, Figure 5 shows that our method exhibits greater resistance to layer collapse.

## 6.3 Other Models on CIFAR10/100 and TinyImageNet

In Figure 4, it is clear that MIPP performs at least as well as baselines, whether PaI or PaT, on most models when CIFAR10, CIFAR100, or TinyImageNet are acting as input. When PaI ResNet18 and ResNet34 at high sparsities MIPP outperforms baselines by over 15%. However, our method is more computationally demanding than PaI competitors, as shown in Appendix G. We observe similarly impressive results when pruning a VGG19 trained on CIFAR100. These results demonstrate that certain global pruning objectives can be used to PaI or PaT. In Figure 5 it is clear that MIPP is consistently the most resistant to layer collapse for all model dataset combinations.

In Figure 6, we observe that MIPP selects highly regularized layer-wise PRs depending on the network structure, particularly under PaI. Notably, for both ResNet34 and 50 MIPP exhibits both inter-block and intra-block patterns. This can be explained by these networks respective structures as shown in Appendix D. This is significant because recent works Pham et al. (2024); Frankle and Carbin (2019) suggest that discovering optimal layer-wise PRs is the sole aspect of PaI that improves performance. In contrast, baseline methods exhibit critical limitations: ProsPr and Grasp tend to induce layer collapse predominantly in the deeper layers, contrasting with other PaI methods which may exhibit collapse in either the initial or final layers. In comparison, PaT baselines generally yield stable layer-wise PR selections. For detailed comparisons, see Appendix F.2.

## 7 CONCLUSION

In this paper, we have introduced MIPP, an activation-based pruning method that can be applied both before and after training. The core principle of MIPP is to remove neurons or filters from layers if they do not transfer entropy to the subsequent layer. Consequently, MIPP preserves MI between the activations of adjacent layers and, therefore, between the data and masks. We have presented a comprehensive performance evaluation of MIPP considering a variety of datasets and models. Our experimental evaluation has demonstrated the effectiveness of MIPP in pruning trained and untrained models of increasing complexity.

## ETHICS STATEMENT

Our investigation into mutual information-preserving neural network pruning presents minimal direct ethical concerns. Since the focus is theoretical in nature and centered on fundamental principles, we do not anticipate significant ethical concerns arising from this research.

## REPRODUCIBILITY STATEMENT

We provided a comprehensive description of the algorithm and required hyperparameters in the main body of the paper and the appendixes. We will also make the code available upon publication.

## LLM USAGE STATEMENT

We employed Large Language Models (LLMs) throughout this work for editorial and technical assistance. LLMs helped improve the clarity of our exposition, transcribe mathematical expressions into LaTeX, enhance figure captions, and implement Python code. They also streamlined our literature review process by generating properly formatted BibTeX entries.

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

# A    NOTATION

Table 1: Summary of Notational Conventions.

| Type | Notation |
|------|----------|
| Loss function | l |
| The function describing layer $l$ | $f_l$ |
| The function describing layer $l$ once masked and re-trained | $h_l$ |
| A vector of activations associated with layer l | $\boldsymbol{x}_l^n$ |
| RV describing the activations of node i in layer l | $X_l^i$ |
| Set of RVs describing all activations in layer l (one realization being $\boldsymbol{x}_l^n$) | $\mathcal{X}_l$ |
| Image vector | $\boldsymbol{x}_0^n$ |
| Set of pixel RVs describing input data (one realization would be an image $\boldsymbol{x}_0^n$) | $\mathcal{X}_0$ |
| RV describing different datasets (one realization would be an dataset such as MNIST or $\mathcal{X}_0$) | $\mathcal{D}$ |
| The mask vector used for layer l | $\boldsymbol{m}_l^n$ |
| The set of masks established for all layers in a model | $\mathcal{M}_0$ |
| The RV describing the different masks established for different data (one realization: $\mathcal{M}_0$) | $\mathcal{M}$ |
| Set of RVs describing pruned activations in layer l | $\mathcal{X}_l'$ |
| RV describing different pruned activations in layer l occurring due to different datasets (realization:$\mathcal{X}_l'$) | $\mathcal{D}_l'$ |
| Weight matrix | $\boldsymbol{W}^{m \times n}$ |
| Bias vector | $\boldsymbol{b}^m$ |
| Activation function | a |

# B    ALGORITHMS

In this section, we present not only the overall MIPP algorithm but also TERC with MI ordering algorithm, which maintains MI between adjacent layers in a network.

---
**Algorithm 1** MIPP.

---
**Input**:    Activations of all layers:    $\mathcal{X}_l$.    **Output**:    $\mathcal{M}_0$ (a desirable set of node masks).

1: Initialize empty set of masks: $\mathcal{M}_0 = \emptyset$.
2: **for** $l \in [1, L]$ **do**
3:     $\mathcal{X}_{l-1}' = $ Algorithm 2$(\mathcal{X}_{l-1}, \mathcal{X}_l)$
4:     **for** $i \in [0, n]$ **do**
5:        $\boldsymbol{m}_{l-1}^n(i) = \begin{cases} 1 & \text{if } X_{l-1}^i \in \mathcal{X}_{l-1}', \\ 0 & \text{otherwise.} \end{cases}$
6:     **end for**
7:     $\mathcal{M}_0 = \mathcal{M}_0 \cup \boldsymbol{m}_{l-1}^n$
8: **end for**
9: **return** $\mathcal{M}_0$

---

# C    PROOF OF THEOREM 1

In this section, we prove Theorem 1. To begin, we present axioms that will be used throughout the proof.

- Firstly, we apply TERC to bijectively preserve MI between activations in layers such that $I(\mathcal{X}_{l-1}'; \mathcal{X}_l) = I(\mathcal{X}_{l-1}; \mathcal{X}_l)$. Given that $\mathcal{X}_l = f(\mathcal{X}_{l-1})$, this implies that from the pruned upstream layer we should be able to perfectly reconstruct the original layer $I(\mathcal{X}_l; \mathcal{X}_{l-1}') = H(\mathcal{X}_{l-1})$ Westphal et al. (2024).

- With probability one can we recover the mask if we have access to the masked activations $p(m|\mathcal{X}_{l-1}') = 1$.

---

**Algorithm 2** TERC with MI ordering.

---

**Input**: Activations of layers $L$ and $L-1$: $\mathcal{X}_L$ and $\mathcal{X}_{L-1}$. **Output**: $\mathcal{X}'_{L-1}$ (a desirable subset of nodes).

1: Initialize $\mathcal{X}'_{L-1} = \text{sort}_{\text{desc}}\left(\mathcal{X}_{L-1}, I(X^i_{L-1}; \mathcal{X}_L)\right)$
2: **for** $X^i_{L-1} \in \mathcal{X}_{L-1}$ **do**
3:   **if** $I(\mathcal{X}'_{L-1}\backslash X^i_{L-1}; \mathcal{X}_L) = I(\mathcal{X}_{L-1}; \mathcal{X}_L)$ **then**
4:     $\mathcal{X}'_{L-1} = \mathcal{X}'_{L-1}\backslash X^i_{L-1}$
5:   **end if**
6: **end for**
7: **return** $\mathcal{X}'_{L-1}$

---

- With probability one can we recover the masked activations if we have access to the full activations $p(\mathcal{X}'_{l-1}|\mathcal{X}_{l-1}) = 1$.
- We assume that all the information in the data is discrete (to overcome the problem described in Saxe et al. (2019) and included in the first layer of the activations $I(\mathcal{X}_0; \mathcal{X}_1) = H(\mathcal{X}_0)$. Finally, for this proof, we also assume a network with one set of activations to prune.

To begin, we remind the reader that $\mathcal{D}$ is a distribution from which we sample input data. Therefore, an instance of $\mathcal{D}$ can be written as the input data to our NN, written $\mathcal{X}_0$. Meanwhile, $\mathcal{M}$ is a RV whose realizations are the sets of masks derived using a pruning method, denoted $\mathcal{M}_0$. We use these observations to complete the proof.

$$I(\mathcal{D}; \mathcal{M}) = \sum_{\mathcal{X}_0 \in \mathcal{D}, \mathcal{M}_0 \in \mathcal{M}} p(\mathcal{X}_0, \mathcal{M}_0) \log\left(\frac{p(\mathcal{X}_0, \mathcal{M}_0)}{p(\mathcal{X}_0)\, p(\mathcal{M}_0)}\right) \tag{2}$$

(substituting in $p(\mathcal{X}_1|\mathcal{X}_0, \mathcal{M}_0) = 1$ we obtain)

$$= \sum_{\mathcal{X}_0 \in \mathcal{D}, \mathcal{M}_0 \in \mathcal{M}} p(\mathcal{X}_0, \mathcal{M}_0) \log\left(\frac{p(\mathcal{X}_0, \mathcal{M}_0, \mathcal{X}_1)}{p(\mathcal{X}_0)\, p(\mathcal{M}_0)}\right) \tag{3}$$

(substituting in $p(\mathcal{X}'_1|\mathcal{X}_0, \mathcal{M}_0, \mathcal{X}_1) = 1$ we obtain)

$$= \sum_{\mathcal{X}_0 \in \mathcal{D}, \mathcal{M}_0 \in \mathcal{M}} p(\mathcal{X}_0, \mathcal{M}_0) \log\left(\frac{p(\mathcal{X}_0, \mathcal{M}_0, \mathcal{X}_1, \mathcal{X}'_1)}{p(\mathcal{X}_0)\, p(\mathcal{M}_0)}\right) \tag{4}$$

$$= \sum_{\mathcal{X}_0 \in \mathcal{D}, \mathcal{M}_0 \in \mathcal{M}} p(\mathcal{X}_0, \mathcal{M}_0) \log\left(\frac{p(\mathcal{M}_0|\mathcal{X}_0, \mathcal{X}_1, \mathcal{X}'_1)p(\mathcal{X}_0, \mathcal{X}_1, \mathcal{X}'_1)}{p(\mathcal{X}_0)\, p(\mathcal{M}_0)}\right) \tag{5}$$

(because $p(\mathcal{M}_0|\mathcal{X}_1, \mathcal{X}'_1) = 1$ we obtain) $\tag{6}$

$$= \sum_{\mathcal{X}_0 \in \mathcal{D}, \mathcal{M}_0 \in \mathcal{M}} p(\mathcal{X}_0, \mathcal{M}_0) \log\left(\frac{p(\mathcal{X}_0, \mathcal{X}_1, \mathcal{X}'_1)}{p(\mathcal{X}_0)\, p(\mathcal{M}_0)}\right) \tag{7}$$

$$= \sum_{\mathcal{X}_0 \in \mathcal{D}, \mathcal{M}_0 \in \mathcal{M}} p(\mathcal{X}_0, \mathcal{M}_0) \log\left(\frac{p(\mathcal{X}'_1|\mathcal{X}_0, \mathcal{X}_1)p(\mathcal{X}_0, \mathcal{X}_1)}{p(\mathcal{X}_0)\, p(\mathcal{M}_0)}\right) \tag{8}$$

(because $p(\mathcal{X}'_1|\mathcal{X}_0, \mathcal{X}_1) = 1$ we obtain)

$$= \sum_{\mathcal{X}_0 \in \mathcal{D}, \mathcal{M}_0 \in \mathcal{M}} p(\mathcal{X}_0, \mathcal{M}_0) \log\left(\frac{p(\mathcal{X}_0, \mathcal{X}_1)}{p(\mathcal{X}_0)\, p(\mathcal{M}_0)}\right) \tag{9}$$

$$= \sum_{\mathcal{X}_0 \in \mathcal{D}, \mathcal{M}_0 \in \mathcal{M}} p(\mathcal{X}_0, \mathcal{M}_0) \log\left(\frac{p(\mathcal{X}_1|\mathcal{X}_0)p(\mathcal{X}_0)}{p(\mathcal{X}_0)\, p(\mathcal{M}_0)}\right) \tag{10}$$

(because $p(\mathcal{X}_1|\mathcal{X}_0) = 1$ we obtain)

$$= \sum_{\mathcal{X}_0 \in \mathcal{D}, \mathcal{M}_0 \in \mathcal{M}} p(\mathcal{X}_0, \mathcal{M}_0) \log\left(\frac{1}{p(\mathcal{M}_0)}\right) \tag{11}$$

$$= H(\mathcal{M}). \tag{12}$$

As previously pointed out, an instance of our masking variable $\mathcal{M}$ is a single set of masks $\mathcal{M}_0$. It is clearly possible to derive this mask from $\mathcal{X}_1'$; therefore, we obtain $\mathcal{M}_0 = \text{f}(\mathcal{X}_1')$ and $H(\mathcal{M}) \leq H(\mathcal{D}_1')$ (where $\mathcal{D}_1'$ is the RV from which $\mathcal{X}_1'$ is sampled).

We can then write:

$$H(\mathcal{M}) \leq H(\mathcal{D}_1') \tag{13}$$

$$\leq - \sum_{\mathcal{X}_1' \in \mathcal{D}_1'} p(\mathcal{X}_1') \log p(\mathcal{X}_1') \tag{14}$$

(because of TERC's bijective MI preservation we can sub in $p(\mathcal{X}_1|\mathcal{X}_1') = 1$)

$$\leq - \sum_{\mathcal{X}_1' \in \mathcal{D}_1'} p(\mathcal{X}_1', \mathcal{X}_1) \log p(\mathcal{X}_1', \mathcal{X}_1) \tag{15}$$

$$\leq - \sum_{\mathcal{X}_1' \in \mathcal{D}_1'} p(\mathcal{X}_1'|\mathcal{X}_1)p(\mathcal{X}_1) \log p(\mathcal{X}_1'|\mathcal{X}_1)p(\mathcal{X}_1) \tag{16}$$

(by repeating the process above but inserting $p(\mathcal{X}_0|\mathcal{X}_1) = 1$)

$$\leq H(\mathcal{D}). \tag{17}$$

We have proven that, if pruning using a method that bijectively preserves MI between pruned and unpruned activations, the upper bound on $I(\mathcal{D}; \mathcal{M})$ can be expressed as $I(\mathcal{D}; \mathcal{M}) \geq H(\mathcal{D})$.

Table 2: Comparison of training parameters across datasets. LR notation: initial learning rate, [epochs at which LR is decayed by 0.1], #total epochs.

| Dataset | MNIST | CIFAR10 | CIFAR100 | TinyImageNet |
|---|---|---|---|---|
| Solver | SGD (0.9, 1e-4) | SGD (0.9, 5e-4) | SGD (0.9, 5e-4) | SGD (0.9, 5e-4) |
| Batch Size | 256 | 256 | 256 | 256 |
| LR | 1e-2, [30,60], #90 | 1e-1, [100,150], #200 | 1e-1, [100,150], #200 | 1e-1, [150,200], #250 |
| LR (re-train) | 1e-2, [30], #60 | 1e-2, [60,90], #120 | 1e-2, [60,90], #120 | 1e-2, [100,150], #200 |

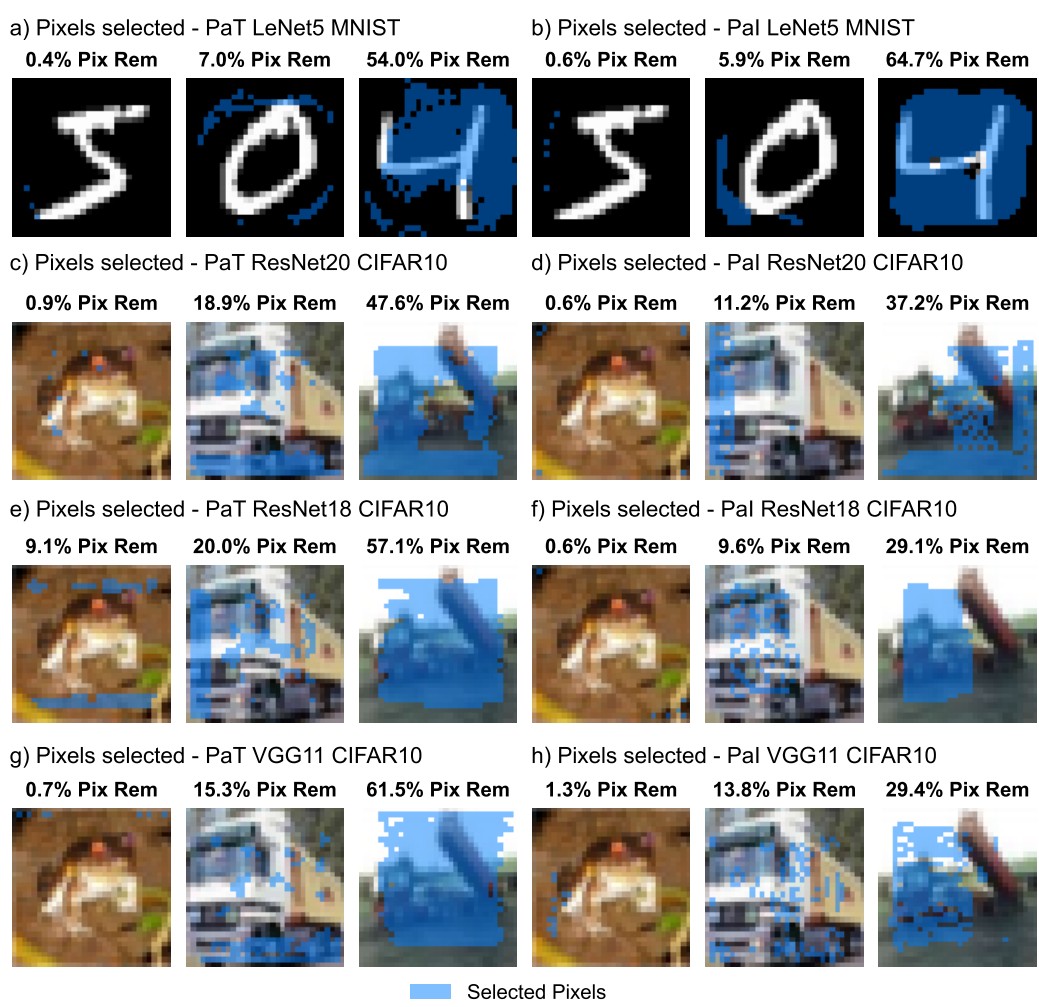

Figure 7: Visual representation of the features selected using MIPP at different sparsities on different models and datasets (blue implies selected).

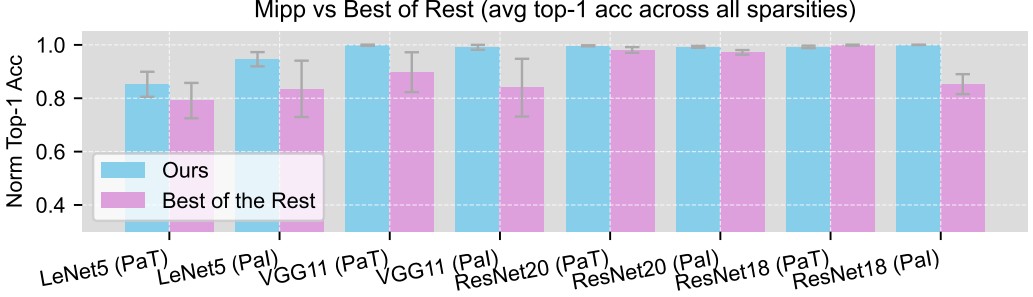

Figure 8: Normalized accuracy of MIPP vs best of the rest when pixel selection occurs.

## D RESNET STRUCTURE

In this section, we present Figure 10, which illustrates the structure of some of the ResNets investigated explaining the per-layer pruning ratios discovered in Figure 9.

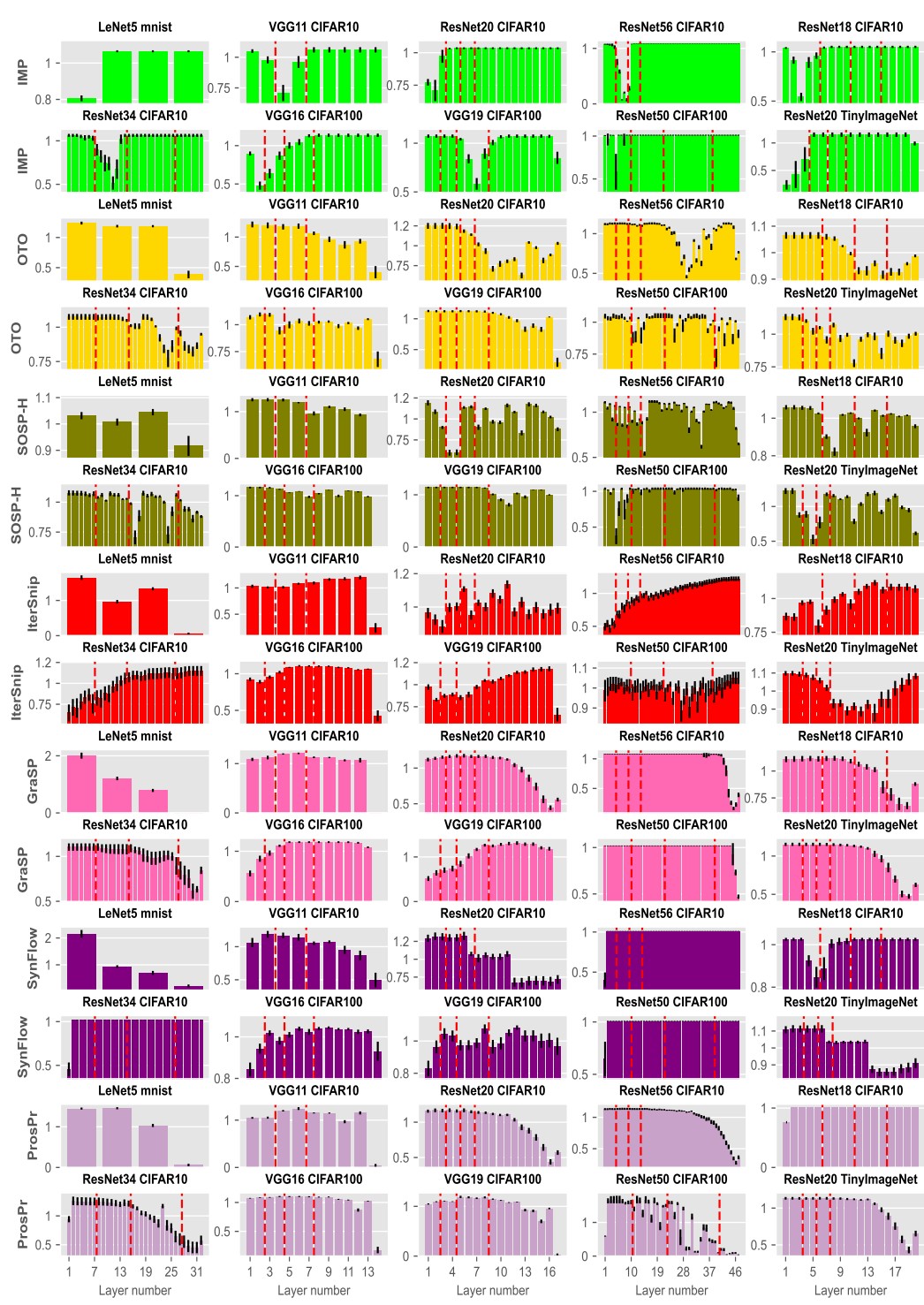

Figure 9: Layer-wise pruning ratios. Normalized by division of the average PR achieved for that run.

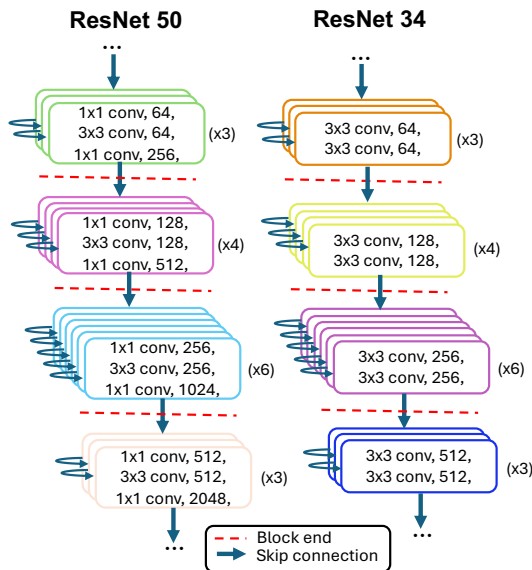

Figure 10: ResNet34 and ResNet50 structures: the architectural elements of the network explain the periodicity of the per-layer PRs derived using our method.

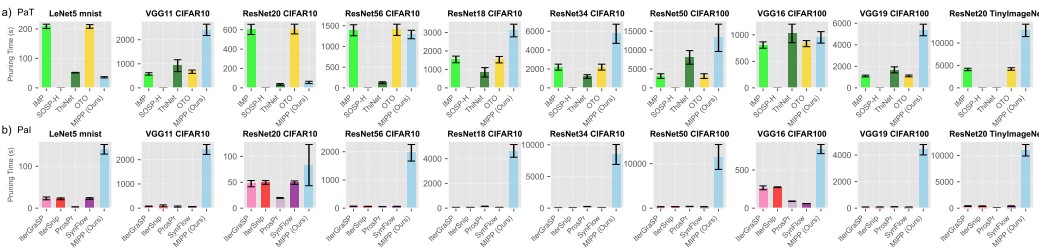

Figure 11: Pruning times across architectures for all experimental runs. Error bars show 95% confidence intervals. MIPP demonstrates competitive timing with other pruning-after-training methods, particularly for narrower architectures.

# E  FURTHER EXPERIMENTAL SETTINGS

## E.1  $I(\mathcal{D}; \mathcal{M})$ VS TOP-1 ACC

In this section, we explain the experimental settings for the results achieved in Figure 2 a). In these simple introductory experiments, we aimed to provide empirical evidence validating the results of Kumar et al. (2024) in different settings. To achieve this, we used synthetic data to generate a mask for which we could calculate $I(\mathcal{D}; \mathcal{M})$, and then applied this mask to all layers of the network. We would then train this network (controlling for the initialized weight matrices) and present the final accuracy seen in the figure.

**Data.** In this case, to simplify the process of deriving our masks $\mathcal{M}_0$, our synthetic data was an $N$-dimensional vector of Bernoulli distributions. Of these $N$ Bernoulli distributions, half were described by $\sim \text{Bernoulli}(0.5)$, while the other half were described by $\sim \text{Bernoulli}(0.999999)$. We denote this vector as $\mathbf{d}^N$. To generate masks with high MI with the data, they should accurately reflect the patterns in the input data. For instance, if we are generating a set of masks with perfect MI with the data, and the informative Bernoulli distributions occupy the first 25 positions, then our mask will have its first 25 positions set to prune. Meanwhile, if the informative Bernoulli distributions appeared once in every other array element, we may repeat this with the positions we mask. What matters for perfect MI is that the masks are a perfect function of the distribution vector, represented

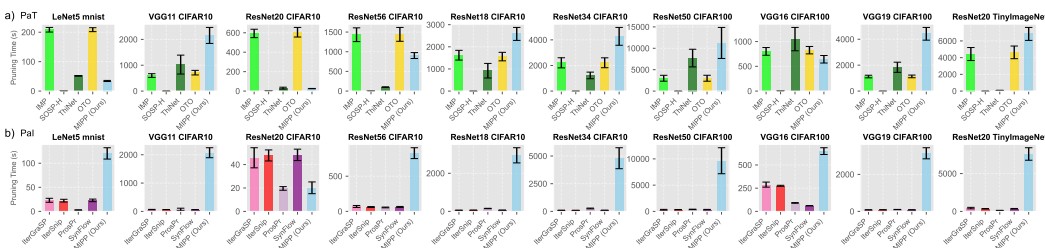

Figure 12: Pruning times for high-sparsity scenarios ( >92% of neurons removed). MIPP's relative performance improves in high-sparsity regimes due to reduced node comparison costs.

as $\mathcal{M}_0 = f(\mathbf{d}^N)$. In our case, this function is simply a 1-to-1 mapping. To reduce MI, we simply add randomness. For instance, if we aimed to reduce the MI by half, we would have only have half our masking vector be dependent on the data, while the rest is random. Finally, for these experiments, the target (i.e., $y$) is a simple sum of all the inputs.

The output data we were trying to predict was a sum of the inputs.

**Models.** For our simple feed-forward neural network (NN), we used 50 inputs and 3 hidden layers, each containing 50 nodes. In contrast, our convolutional neural network (ConvNet) had one convolutional layer with 1 input channel, 8 output channels, and a kernel size of 3. We then flattened the output and fed it into a linear network with 384 units.

### E.2 What PaI methods maximize $I(\mathcal{D}; \mathcal{M})$?

In this section, we explain the experimental settings for the results reported in Figure 2 b). The goal of these experiments is to show that for a simple synthetic network characterized by redundancies, MIPP establishes masks that have a greater MI with the data. We employed a simple MLP composed of 10 hidden nodes, with input data that was also a vector of dimension 10, where, similarly to in Appendix E.1, this vector was made of samples from Bernoulli distributions, half of which were informative ($\sim$ Bernoulli$(0.5)$) while the other half ($\sim$ Bernoulli$(0.999999)$) were not. The network takes this input vector of zeroes and ones and converts them from a binary value to a decimal one. In this case, a perfect network will have a weight matrix that has powers of two along the diagonal. We then use multiple PaI methods to establish pruning masks. To calculate $I(\mathcal{D}; \mathcal{M})$, we use the methods described in Covert et al. (2020) with a small network of two hidden layers with 50 nodes and BCE loss.

We observe that only MIPP can define pruning masks that are data-dependent. This is due to the following reasons: firstly, the network performs perfectly and there is no loss, so the derivative-based metrics are zero; and, secondly, the activations give perfectly redundant information, masking them would have the exact same effect on the loss (assuming you use a cross entropy loss and not MSE). Under such conditions, static ranking methods produce results that resemble randomness.

### E.3 Data Augmentation Techniques

For the CIFAR-10 dataset, we applied standard data augmentation techniques, which included random cropping with padding and random horizontal flipping. These augmentations are commonly used to enhance model generalization by introducing variations in the training data. In the case of CIFAR-100, we employed additional augmentation methods beyond the standard techniques. Specifically, we used mixup (Zhang et al., 2018), which creates virtual training examples by combining pairs of images and their labels, and cutout (DeVries and Taylor, 2017), which randomly masks out square regions of an image to simulate occlusion and encourage the network to focus on more distributed features. These advanced techniques were included to further enhance performance due to the increased complexity of the CIFAR-100 dataset. For TinyImageNet, we again adopted a standard suite of transforms. The transformations included RandomResizedCrop with a scale range of 0.8 to 1.0, which crops and resizes images more variably than standard cropping, alongside random horizontal flipping. To further diversify the training data, we introduced ColorJitter to perturb brightness, contrast, saturation, and hue, as well as small random rotations of up to 10 degrees. The images were then normalized

using TinyImageNet-specific statistics. Additionally, we applied RandomErasing (similar to cutout) with a probability of 0.5 to randomly mask small regions of the image, further encouraging robust feature learning.

### E.4 HYPERPARAMETERS

#### E.4.1 VISION TRAINING AND RE-TRAINING

Table 2 contains a comparison of the training parameters used for the vision training and retraining across datasets.

#### E.4.2 PRUNING

As explained in the main paper, our method involves masking features and re-training a layer to check if the loss decreases below the original level. For the re-training steps once the mask has been applied we use 100 epochs with 20 mini-batches of activations with early stopping. At the start of our algorithm, we also rank the features based of their MI. For this calculation, we again use the same layer for 5 epochs with the same 20 mini-batches of activations (i.e., $20 \times$ batch size samples). We use our method to prune all linear and convolutional layers. We prune the batch-normalization nodes associated with nodes in linear/convolutional layers, while skip connections in ResNets remain unaffected. We note that the batch-normalization layer can be considered as part of the function $f_{l-1}$ and therefore retrained to form $h_{l-1}$. Consequently, MIPP considers batch-normalization layers to greater effect than other pruning methods. For OTO and IMP, we used 10 iterations with 20 retraining epochs. The learning rates are reported in Table 2.

## F FURTHER EXPERIMENTS

### F.1 FEATURE SELECTION EXPERIMENTS

In this section, we investigate MIPP's ability to select features. In particular, we examine the pixels it identifies from the MNIST and CIFAR10 datasets. MIPP selects features in the exact same manner it selects nodes, i.e., by verifying whether entropy is transferred from the pixels to the nodes of the first layer. In order to do this, we extract the first layer and rank the pixels based on their impact on reconstructing the activations of the next layer, from least to most significant. We then sequentially evaluate the pixels in this order, testing whether retraining the layer with each pixel masked results in a loss that is lower than or equal to the loss obtained using all the pixels.

In Figure 7, we observe a clear preference for selecting central pixels over those located at the periphery, which are generally less informative. In panels (a) and (b) of Figure 7, we present results that align with previous research on MNIST pixel importance Covert et al. (2020). These studies have demonstrated that the most informative pixels are typically concentrated in the center of the images, with a slight bias toward the right side. Additionally, an intuitive selection pattern is evident in the CIFAR-10 dataset. When pixel selection is performed after training, the selected pixels exhibit a high degree of uniformity. In contrast, when selection occurs at initialization, regular patterns still emerge, although the spacing between selected pixels is less consistent.

In Figure 8, we present the average accuracy achieved when we prune models using MIPP and our baselines. Unlike the experiment reported in the main body of the paper, we have also used MIPP to select pixels. In both figures, we observe that MIPP outperforms the baselines. This is because, unlike any of the baselines, the features are selected in a manner that is dependent on the pruned model. MIPP can compress both features and the underlying model simultaneously such that the results are compatible, preventing ML practitioners from having to use different methods for feature and model compression. Often, combining compressed input and compressed models can lead to performance degradation.

### F.2 LAYER WISE PRUNING RATIOS ESTABLISHED USING OTHER METHODS

In the main paper, we present the per-layer PRs obtained by MIPP. In Figure 6, we present these results for the other methods taken into consideration.

# G  COMPUTATIONAL COMPLEXITY ANALYSIS

In this section, we analyze the computational complexity and runtime of MIPP, which consists of two distinct stages: (1) mutual information (MI)-based node ordering and (2) entropy-conserving node pruning.

## G.1  THEORETICAL COMPLEXITY

The first stage involves evaluating nodes in a given layer by sequentially masking all but one node, performing brief layer-wise retraining, and measuring each node's contribution to reconstructing subsequent activations. This MI-ordering phase exhibits a complexity of $\mathcal{O}(n)$, where $n$ is the number of nodes in the layer, as each node requires individual evaluation followed by pairwise comparison.

The second stage consists in removing nodes that do not transfer significant entropy to the next layer. his phase has an effective complexity of $\mathcal{O}(k)$, where $k$ denotes the number of nodes retained after pruning. Since nodes with minimal impact can be identified without requiring full retraining, the process remains computationally efficient.

Combining both stages, the per-layer complexity of MIPP is $\mathcal{O}(nk)$. For a network of depth $d$ with comparable layer widths, the total complexity becomes $\mathcal{O}(d(nk))$. This scaling makes MIPP competitive among pruning-after-training (PaT) methods while remaining more computationally intensive than pruning-at-initialization (PaI) approaches.

## G.2  EMPIRICAL PERFORMANCE

Figure 11 demonstrates MIPP's empirical runtime characteristics. The non-linear relationship between network width and pruning time is evident, with particularly favorable scaling for narrower architectures like ResNet-20 compared to wider networks like VGG-19. This aligns with our theoretical complexity analysis, where the $n^2$ term dominates for layers with many nodes.

Notably, MIPP's computational cost scales advantageously with pruning severity. As shown in Figure 12, when pruning to ultra-high sparsity levels (removing >92% of neurons), MIPP's relative performance improves compared to moderate sparsity scenarios (Figure 11). This occurs because high-sparsity pruning reduces both the $n^2$ ordering cost (fewer nodes to compare) and the $k$ retention cost simultaneously.

While MIPP's absolute runtime exceeds standard PaI methods (as visible in both subfigures), it provides unique advantages for edge deployment: the method produces compact models with greater on-device learning capacity than conventional PaI approaches. This makes MIPP particularly valuable for applications requiring continuous edge learning, such as autonomous systems or IoT devices, where the computational cost is justified by the resulting model quality and adaptability.

# H  DISCUSSION OF RESULTS IN SAXE ET AL. (2019)

In theory, the mutual information $I(\mathcal{X}^l; \mathcal{X}^{l-1})$ between adjacent layers of a neural network is infinite, as demonstrated by Saxe et al. (2019). This occurs because there exists an exact deterministic mapping from layer $\mathcal{X}^{l-1}$ to $\mathcal{X}^l$ in an idealized noise-free network. The deterministic nature of this transformation would imply zero conditional entropy $H(\mathcal{X}^l|\mathcal{X}^{l-1}) = 0$, leading to unbounded mutual information $I(\mathcal{X}^l; \mathcal{X}^{l-1}) \rightarrow \infty$.

However, in practice we estimate this quantity using the method from Covert et al. (2020), which yields a finite, approximate lower bound rather than the theoretical infinite value. This approximation arises from two fundamental limitations. First, the estimator only becomes exact when implemented with a binary cross-entropy loss $\mathcal{L}_{\text{BCE}}$, which is impossible to achieve in our continuous activation setting where $\mathcal{X}^l \in \mathbb{R}^d$. Second, all practical training pipelines introduce various sources of randomness - including stochastic initialization, dropout with rate $p_{\text{drop}}$, mini-batch sampling $\mathcal{B}_t \subset \mathcal{D}$, and label smoothing $\alpha$ - that broaden the conditional distribution $p(\mathcal{X}^l|\mathcal{X}^{l-1})$ from a Dirac delta $\delta_{f(x)}$ to one with finite entropy $H(\mathcal{X}^l|\mathcal{X}^{l-1}) > 0$.

Consequently, the mutual information values we report are empirically measurable, i.e., practical quantities that satisfy $0 < \hat{I}(\mathcal{X}^l; \mathcal{X}^{l-1}) < \infty$ under real-world training conditions, rather than the idealized $I(\mathcal{X}^l; \mathcal{X}^{l-1}) \to \infty$ that would emerge from a perfect deterministic mapping $f: \mathcal{X}^{l-1} \to \mathcal{X}^l$ (note that in the paper we refer to the MI with randomness as $I(X;Y)$ rather than $\hat{I}(X;Y)$ for clarity of presentation). This distinction is crucial for a correct interpretation of our results.

## I How does $I(M; \mathcal{D})$ relate to generalization?

Through Figure 2 b) and Theorem 1 we have demonstrated that the application of MIPP results in higher values of $I(M; \mathcal{D})$. However, Kumar et al. related this quantity to train, not test, loss. This leaves a gap between our method that aims to raise $I(M; \mathcal{D})$ and the metric we use to measure performance: test-time accuracy.[3]

To address this gap, we note that Kumar et al.'s theorem can be interpreted as follows: for a fixed training loss below the noise level, increasing the effective parameter count (including $I(M; \mathcal{D})$) allows one to achieve the same training loss with a smoother function. Function smoothness, as measured by the Lipschitz constant, has been shown to be directly related to generalization bounds in numerous contexts. For instance, Bartlett et al. (2017) and Neyshabur et al. (2017) demonstrated that generalization bounds can be expressed in terms of the product of layer spectral norms rather than parameter count. The Lipschitz constant is upwardly bounded by the product of layer-wise spectral norms. Therefore, smaller Lipschitz constants should enhance generalizability in deep nets according to Neyshabur et al. (2017) and Bartlett et al. (2017), provided we control for both margins and structural capacity.

However, even with preserved margins, the relationship between $I(M; \mathcal{D})$ and generalization remains nuanced, as the theoretical frameworks of Bartlett et al. (2017) and Neyshabur et al. (2017) do not consider the following: increasing $I(M; \mathcal{D})$ could harm generalization if our masks overfit to noise rather than capturing meaningful structure. This concern aligns with the line of work initiated by Xu and Raginsky (2017), which demonstrated that generalization error can be bounded by the MI between the final hypothesis $h$ and data generating processes $\mathcal{D}$. Given that our mask $M$ can be recovered from our final weights $W$, it is possible to relate these two quantities via a data processing inequality $I(W; \mathcal{D}) \geq I(M; \mathcal{D})$. However, this bound is vacuous as the final weights can carry vastly more entropy than the masks. This creates an apparent tension: while Kumar et al.'s framework suggests higher $I(M; \mathcal{D})$ enables smoother interpolation, Xu and Raginsky (2017) warn that excessive $I(h; \mathcal{D})$ indicates memorization of training data specifics. We believe MIPP resolves this tension due to the constraints of structural pruning. While the masks increase $I(M; \mathcal{D})$ to capture meaningful patterns, they also reduce the effective capacity of the network, preventing the growth of $I(W; \mathcal{D})$ that would lead to overfitting.

## J Training Times Across Heterogeneous GPU Resources

Training durations for the evaluated models varied considerably due to the heterogeneous GPU configurations within the HPC cluster, which included nodes equipped with GPUs ranging from NVIDIA P100 (16 GB, Pascal) to A100 (80 GB, Ampere). For the following datasets, the observed wall-clock time per epoch spanned the following ranges:

- **MNIST (28×28)**: LeNet-5: ~10–60 seconds per epoch.
- **CIFAR-10 (32×32)**:
    - VGG11: ~2–5 minutes per epoch
    - ResNet20: ~1.5–4 minutes per epoch
    - ResNet56: ~3–6 minutes per epoch
    - ResNet18: ~4–8 minutes per epoch
    - ResNet34: ~5–10 minutes per epoch
- **CIFAR-100 (32×32)**:

---

[3]Note that the results in Figure 2 b) are computed on toy models, as obtaining unbiased estimates of $I(M; \mathcal{D})$ for larger networks would require generating prohibitively large numbers of pruning masks.

- ResNet50: ~6–18 minutes per epoch
- VGG16: ~8–14 minutes per epoch
- VGG19: ~9–16 minutes per epoch

- **Tiny ImageNet (64×64):**

  - ResNet20: ~4–12 minutes per epoch

These variations reflect both the computational complexity of the models and the diversity of GPU hardware used during training.

