# OpenReview forum: "Mutual Information Preserving Neural Network Pruning"
_ICLR.cc/2026/Conference — Submitted to ICLR 2026_

### Official Review · Reviewer_z8gy · 2025-10-30

**Soundness:** 3
**Presentation:** 3
**Contribution:** 2
**Rating:** 4
**Confidence:** 3

**Summary:**

This paper proposes a novel neural network pruning method termed Mutual Information Preserving Pruning. The core innovation lies in achieving efficient structural pruning by preserving mutual information between activations of adjacent layers, applicable in both pre-training and post-training scenarios.

**Strengths:**

The dynamic pruning method proposed in this paper, which considers mutual information retention, is innovative. Theoretically, it demonstrates that the MI approach can preserve retraining capability while maximizing mutual information between the dataset and the mask. Experiments conducted on ResNet models of varying sizes across datasets including MNIST, CIFAR10/100, and TinyImageNet demonstrate that preserving mutual information yields effective pruning results to a certain extent. The authors also account for both pre-training and post-training pruning scenarios.

**Weaknesses:**

Although the paper presents intriguing perspectives, several critical issues remain unaddressed:
1. The selected models and datasets are overly simplistic. The authors only tested on basic architectures like ResNet, LeNet, and VGG, while datasets were limited to MNIST, CIFAR, and TinyImageNet. This fails to demonstrate the applicability of such methods to current mainstream models (e.g., ViT) and more complex datasets (e.g., ImageNet-1K).
 2. The paper's baselines are outdated, primarily relying on work published between 2020 and 2022. Recent structured pruning approaches like LLMPruner and SliceGPT demonstrate strong applicability to architectures like ViT. The paper lacks comparisons with these methods and fails to extend its analysis to more complex architectures.
 3. The presentation of results is not sufficiently intuitive. All findings are displayed using images, whereas tabular formats would better highlight the method's strengths and weaknesses.

**Questions:**

Refer to Weakness

---

> ### Author Response · Authors · 2025-11-28
> **Response**
>
> We thank the Reviewer for the detailed and very useful comments. In the following we address each concern raised in the review.
>
> ### Comment 1
> **Reviewer:**
> The selected models and datasets are overly simplistic (ResNet, LeNet, VGG on MNIST, CIFAR, TinyImageNet). This limits the demonstration of applicability to modern architectures such as ViT or larger datasets like ImageNet-1K.
>
> **Response:**
> We appreciate this concern. Our choice of models and datasets was driven by computational resource constraints, which prevented us from training or pruning substantially larger architectures.
>
>
> ---
>
> ### Comment 2
> **Reviewer:**
> The baselines are outdated, mostly from 2020–2022. More recent structured pruning methods (e.g., LLMPruner, SliceGPT) demonstrate strong results on ViT-style architectures, yet these comparisons are absent.
>
> **Response:**
> We believe we have selected an appropriate set of baselines for vision-specific structural pruning. If the Reviewer feel there are important works we have overlooked, we would be grateful for your suggestions.
>
> ---

---

### Official Review · Reviewer_huq5 · 2025-10-30

**Soundness:** 2
**Presentation:** 2
**Contribution:** 2
**Rating:** 2
**Confidence:** 3

**Summary:**

The paper utilizes mutual information between layers, starting from the output layer to the input layer, to create a method of structured pruning that can be applied at initialization and after training. This is explored across 4 datasets, namely, MNIST, CIFAR10, CIFAR100, and TinyImageNet, with results for pruning at initialization (PaI) and after training (PaT).

**Strengths:**

- Provides theoretical justification for the design of the method.

- The method is explained well.

- A large selection of datasets is explored.

- Benchmarked against a collection of pruning methods, namely: IMP (PaT), SOSP-H (PaT), OTO (PaT), IterGraSP (PaI), IterSnip (PaI), ProsPR (PaI), and SynFlow (PaI), resulting in 3 for PaT and 4 for PaI.

**Weaknesses:**

Although the method allows for pruning at higher ratios while maintaining performance, compared to the methods explored, it suffers from a high discovery cost.

In Figure 2a. Is it the train or test accuracy being presented? Additionally, it states that increasing the mutual information results in a performance increase; however, between an $I(\mathcal{D}; \mathcal{M})$ of 10 and 15, there is a performance drop for the conv model, with no explanation given.

Figure 2c does not state the `different sparsity ratios` used to form the figure. What are these ratios that have been averaged? It would be more informative to provide a table of the sparsity ratio and the resulting change in performance from the baseline. In addition, is the `best of rest` selected at each sparsity ratio or the best mean method across all sparsity ratios?

On line `098` , it states `To illustrate this visually, in Figure 2.b we show that only MIPP can derive useful pruning masks for a synthetic NN characterized by nodes sharing redundant information`, however, Figure 2.b figure does not show that it can derive **useful** masks. It shows it can derive masks. As the performance is not shown.

On line 217 , it states `In other words, MIPP ensures re-trainability`; however, this is not the case, as shown with Figure 5, the `LeNet5 on MNIST` PaT and PaI, `VGG11 on CIFAR10` PaT, `VGG16 on CIFAR100` PaT, and `VGG19 on CIFAR100` PaT and PaI show that the MIPP method results in layer collapse, thus the method does not ensure re-trainability.

The paper throughout states the `average` but does not state which average is used: mean, median, or mode?  In-addiiton it does not state how many runs were used to form said average. Furthermore more what do the error bars represent? Standard Deviation (STD), Standard Error of the Mean (SEM), Min Max?  If STD or SEM, what by $\pm1$ or $\pm2$, etc.

In Appendix F.4.2, it is stated, For the re-training steps once the mask has been applied we use 100 epochs with 20 batches of activations with early stopping  What does 20 batches of activations mean? Does this mean 20 batches of the mini-batches, so for the MNIST task, this is 256, so a total of 5120 activations? If so,  this needs to be made clearer within the paper.

The Prune after Training (PaT) results are when the model is then re-trained for 66.67%, 60%, 60% and 80% for MNIST, CIFAR10, CIFAR100, and Tinyimagent of the original training epochs. Given the large amount of re-training epochs, this can be viewed as a special case of the Prune at Initialisation case, as the model is re-trained for such a significant amount of time. This also does not make a computational equivalent comparison to other methods, as the MIPP Method can, in cases, take a considerable amount of time to compute the pruning mask.

To say this results in not controlling for the computational budget. In addition, re-training for this amount of time introduces potential biases. Why are the hyperparameters selected? Why that number of epochs? There is a lack of justification for the selection.  Do other methods perform better than MIPP with different hyperparameters? This serves as a large weakness in the paper.

## MISC

Figure 3 is unclear. To make this clear, can the x and dots have a color to reflect the deformation applied? Also, is it the train or test accuracy being reported?

Figure 6 has a collection of red lines on the subplots with no explanation. What are the red lines?

Table 2 is unclear what does LR 1e-2 [30,60], #90  mean? I presume this means using a learning rate of 1e-2 at epochs 30 through 60 and then training for a total of 90 epochs?

The appendix figures are not in the correct section-which makes following the appendix difficult.

On line 274, it states `function approximated via loss 1`. What does the `1` represent? Is it a typo?

`MACS` is not defined within the paper.

The citation `Scott E. Fahlman and Christian Lebiere. The cascade-correlation learning architecture. NeurIPS’90, 1990.` has the wrong year; it is 1989, see: https://proceedings.neurips.cc/paper_files/paper/1989/hash/69adc1e107f7f7d035d7baf04342e1ca-Abstract.html. The citation also appears out of place with no real relevance, given that it is used to justify using prime ($^\prime$) notation, which is used in a standard way.

**Questions:**

See weakness and more concretely, the following:

As mentioned as weakness re-training the model after pruning introduces potential areas for biases in the performance of the method. Therefore, can the authors provide One-Shot (With no-retraining) pruning performance on a trained model of the MIPP Method against the benchmarked methods across the datasets explored? This would provide a fairer comparison of the methods and provide insight into the MIPP method and how effective utilizing mutual information is in this manner.

How effective is this method when varying the amount of data to train the MI model? Can the method be sped up by reducing the number of activations trained on? Is there a relationship between performance preservation and the amount of data used to estimate the mutual information? This is asked as for MNIST, 20 batches of 256 activations is a (5120/60000) =  8.53\% of the training data, for CIFAR-10 and 100, this (5120/50000) 10.24% and for TinyImagenet, this is (5120/100000) = 5.12%.

---

> ### Author Response · Authors · 2025-11-28
> **Response**
>
> We thank the Reviewer for the detailed and very useful comments. In the following we address each concern raised in the review.
>
> ### Comment 1
> **Reviewer:**
> In Figure 2a, is this train or test accuracy? Also, you state that increasing mutual information increases performance, but between α = 10 and 15 the Conv model performance drops with no explanation.
>
> **Response:**
> Figure 2a reports **test accuracy**. The small dip between $\alpha$ = 10 and $\alpha$ = 15 is due to normal experimental noise.
>
> ---
>
> ### Comment 2
> **Reviewer:**
> Figure 2c does not state the sparsity ratios used. What ratios were averaged? A table of sparsity ratio vs. performance change would be more informative. Also, is the “best of rest” chosen per sparsity ratio or based on mean performance across all ratios?
>
> **Response:**
> Figure 2c presents **averages across all sparsity ratios**, for both MIPP and “best of rest.”
>
> ---
>
> ### Comment 3
> **Reviewer:**
> Line 098 claims Figure 2b shows that only MIPP can derive “useful” pruning masks, but the figure shows masks only, not performance. How is “useful” defined?
>
> **Response:**
> By “useful”, we meant that MIPP produces masks with **high mutual information between mask and data**, not that performance is displayed. We have revised the wording too: “To illustrate this visually, Figure 2.b shows that only MIPP can derive pruning masks that have non-zero mutual information with the data for a synthetic neural network in which nodes share redundant information.”
>
> ---
>
> ### Comment 4
> **Reviewer:**
> Line 217 states that MIPP “ensures retrainability,” but several cases in Figure 5 demonstrate layer collapse. Therefore, retrainability is not ensured.
>
> **Response:**
> Our intention was to state that retrainability follows under certain theoretical assumptions, which do not always hold in practice. We have revised the manuscript to ensure that this claim is not overstated.
> ---
>
> ### Comment 5
> **Reviewer:**
> The paper repeatedly references averages but never states whether these are mean, median, or mode. It also does not specify how many runs were performed or what the error bars represent (STD, SEM, min–max, CI, etc.).
>
> **Response:**
> We use the **mean** across runs and report **95% confidence intervals**. We will explicitly state the number of runs and define the error bars in the text.
>
> ---
>
> ### Comment 6
> **Reviewer:**
> Appendix F.4.2 mentions using “20 batches of activations,” which is unclear. Does this mean 20 mini-batches? For MNIST, this would be 20 × 256 = 5120 activations.
>
> **Response:**
> Correct, this refers to **20 mini-batches**, each of size 256 for MNIST (5120 activations total).
>
> ---
>
> ### Comment 7
> **Reviewer:**
> Figure 3 is unclear. Can the markers be colour-coded to reflect deformation type? Also, does it show train or test accuracy?
>
> **Response:**
> Figure 3 shows **test accuracy**. We included multiple deformation types not to explore different deformation strengths, but to ensure we did not overfit to the particular randomness introduced by any single deformation. We will make this clearer in the final version of the manuscript.
> ---
>
> ### Comment 8
> **Reviewer:**
> Figure 6 contains unexplained red lines. What do they represent?
>
> **Response:**
> The red lines denote the **ends of blocks** within the network. We have added this to the explanation to the caption.
>
> ---
>
> ### Comment 9
> **Reviewer:**
> Table 2 includes entries such as “LR 1e−2 [30, 60], #90,” which is unclear. Does this mean a learning rate of 1e−2 from epochs 30 to 60 and a total of 90 epochs?
>
> **Response:**
> This denotes a schedule where the learning rate starts at **1e−2** and is decayed by **0.1 at epochs 30 and 60**, with **90 total epochs**.
>
> ---
>
> ### Comment 10
> **Reviewer:**
> Appendix figures appear in the wrong subsections, making the appendix difficult to follow.
>
> **Response:**
> We have re-organised the appendix and place all figures in their correct subsections.
>
> ---
>
> ### Comment 11
> **Reviewer:**
> Line 274 refers to “loss 1.” Is this a typo?
>
> **Response:**
> This refers to **Equation (1)**. We have updated the corresponding section indicating “Equation 1.”
>
> ---

---

### Official Review · Reviewer_Y8wy · 2025-11-05

**Soundness:** 3
**Presentation:** 2
**Contribution:** 3
**Rating:** 2
**Confidence:** 4

**Summary:**

This paper introduces a layerwise structured pruning method which tries to ensure that each pruned set of activations can still reconstruct, fairly well, the following layer. It motivates this with mutual information arguments. It performs experiments in many setting including MNIST, and with many algorithms, including LeNet and VGG, and reports results against a variety of other pruning methods.

**Strengths:**

The overall paper is sensibly motivated in term on mutual information maintenance, though there is some question as to why inter-layer MI matters when the only info we care about is the info at the final layer. The paper proposes a layerwise pruning method that may differ from previous work.

**Weaknesses:**

Writing

The paper is fairly well written, but would really benefit from:

(a) concise obvious statement of the problem (this is fairly good but could be tighter) - really early on say what you want to achieve in non-ambiguous terms; pruning can mean many things an unstructured pruning is broken, so make sure you get it early you are doing structured pruning. Define the task precisely.
(b) a precise statement of the insight. Again, it is good. You get across the insight, in that you say quite a few times you wish to preserve mutual information between activations of adjacent layers. But preserve between masked and unmasked? masked and masked? Elementwise or joint? What do you leverage to make this possible - what do you use for estimating MI? It takes a long while for this to be made clear to the reader (and in some cases it never does). Again get this in early, precisely and concisely. Figure 1 is not so helpful as I haven't been told the notation when I look at it. Get all the info into the caption.
(c) A statement as to why this is not already done? What is the failure mode you are addressing?
(d) A clear statement of the precise method in all its detail. What you actually do. What are your implementational choices? Make this somewhere easy to find. An algorithm perhaps, or a clear and obvious "master" equation, with everything sufficiently defined. As a reader I want to quickly see the purpose, the point, and what you actually do, and then read the paper in more detail filling in any gaps I have.

The paper insight is mostly sound. Borrowing from Tishby (which IMO is helpful, albeit overstated)  a trained neural network has already learnt to compress somewhat given its structure. Pruning just applies a further compression, but in each layer we only want to compress information that gets lost anyway, or perhaps has even already been compressed out by training, but is encapsulated in a known structured reduction in weights or computation.

Though sound, and though I am always happy to hear sensible thinking reiterated, this is not new insight, and so a lot of the question is what does this paper achieve this beyond earlier approaches, and why is this not already done? Beyond that, though, there is one fundamental issue unaddressed: in the context of pruning why is layerwise MI maintenance important? After all the only info you need is information about the final layer. So can't we just focus on MI(X_L,Y), with Y the output layer, which is what most other methods do. So isn't this making life hard for no tangible practical or conceptual gain?

Please could you comment on the other aspect of the mapping. Maintaining information is necessary not sufficient. It is not enough to maintain information; one must maintain representation. Permuting all the nodes in the layer without permuting parameters retains information, but completely destroys network functionality. You cannot escape from commenting on this aspect.

The Theorems should be dropped. Under some approach of making it more precise, Theorem 1 is a statement of the obvious (assuming F is some formalisation of all possible maps from one space to another), and is meaningless and unformulated without statement of what space F is. It is already a well known consequence of the definition of mutual information and the data processing inequality. However it is not precisely defined. The theorem makes a statement about an individual activation, but does not follow in this setting, where a generally-lossy transformation may not be lossy in an individual circumstance. The theorem is stated is wrong: it is really meant to be a statement in expectation in some way. The proof is also imprecise, missing parts and hence is basically wrong.  The theorem also seems of little use. Suppose X′_ {L−1} is a prime factor encrypted representation of X_ {L−1}. Such a mapping is invertible once the encryption key is discovered by prime factorisation. So X′_ {L−1} fully maintains the info about X_{L} and so hence a g exists, but it is completely unattainable without first factorising a vast prime number. This is a pathological example but it illustrates how the theorem provides no tangible value.

I'd simply ditch the Theorem. It is orthogonal to the paper, and seems to be the result of a common but entirely erroneous feeling in some quarters that a paper needs to have a theorem in it. The space could be better used for explaining other things better.

Theorem 2 is also a restatement of the Data Processing Inequality. I really think we should discourage research papers from containing (often imprecise) proofs of fundamental well-known results of Information Theory that have been around for the last 75 years. Again Theorem 2 isn't really used. It too can go.

Previous work:

The key question is how this approach differs from previous work. Information maximisation-based pruning in various guises has been around for a while.

Bin Dai's 2018 paper, Compressing Neural Networks using the Variational Information Bottleneck very much applies optimal information preservation (via an information bottleneck) to the pruning task. Isik presented "An Information-Theoretic Justification for Model Pruning" summarising pruning in very similar terms to how it was described here. In the 1990s, Information theoretic preservation criteria were used for selecting neural network features (which is really the same as activations) - there were many such papers, but the already referenced 1994 paper by Battiti on Using Mutual Information for Selecting Features in Supervised Neural Net Learning is a good example. I think relating stronger to this earlier work would be very good.

The paper would therefore benefit from (a) a deeper look back at the substantive literature in this area and (b) distinguishing the methods here from other methods.

Method:

I do not understand the requirement I(Xl−1; Xl) = I(Xl−1\Xi l ; Xl). This is a really strong requirement. Does it really hold ever? The fact is that you don't use it, because you cannot calculate mutual information, and the estimators you use are noisy and biased (there is no unbiased estimator of MI), and because you introduce a threshold so you never get equality. Hence there is a discrepancy between what you describe and what you actually do.

So what do you actually do? Well here I get a bit worried, as despite all this talk of MI, as far as I can see you just track a local loss, the same way many other pruning methods do? I say "as far as I can see" as equation (1) contains the term $h_{l-1}$ (defined far too late) which involves training, which is happens wrt a specific but unspecified loss.

The problem with talk about mutual information is that getting good mutual information estimators is very hard. But it is not enough to spend ages motivating mutual information (and indeed attempt proving theorems on it!) and then mostly ditch it in favour of something simpler.

The stated mutual information approximation is too cryptic: I(X; Y ) ≈ E[l(f_\theta(\theta), Y )] − E[l(f_X (X), Y )], where most elements of this are undefined( e.g. \theta, both f's and what the expectation is over) and a further cryptic "here each f is some function approximated via loss l" is stated, that again I cannot parse, as the loss l and f are non-commensurate terms that take arguments in different spaces, and so it is not meaningful for f to be approximated by l. Indeed this is generally not close to the MI for many l and f, and is not a good estimator of mutual information in general, and I suspect it is in general, even if  properly defined, it is hard to imagine it can be a great basis for an MI estimator. Its value is all hidden in the unspecified loss, the function choices, and I suspect whatever loss is used is going to be problematic.

Importantly the expectation cannot be computed. We only have finite data. Any estimator based on this expectation is going to be problematic for MI estimation. This issue is unaddressed.

The statement "If the target is discrete, and a cross entropy loss is used, then this value is exactly equal to the ground truth MI" does not seem to be correct as it relies on a function map f for a relationship that might not be a function, but it is hard to know without \theta f_theta f_X and l being defined.

So really AFAICS the mutual information estimator is unclear and the two parts of Eq. (1) appear to not be strongly related.

At the moment the link between all the discussion of mutual information, and what the paper actually does is somewhat lost on me. This is something that needs more care and space. I am afraid as it stands it seems to me that this paper just does pruning as every other layerwise pruning paper does pruning based on some simple prediction error. I really don't see the the difference.

After all this, I am afraid I feel the large run in on the paper is rather disconnected with what is actually done. And what is actually done is the not particularly advantageous to any other pruning method: in the end, the method simply kills activations if they doesn't make things worse in terms of constructing the original downstream activations, potentially after fine-tuning. And the focus on layerwise optimization seems something of a red-herring. In fact other papers have found this to be more problematic, as a layerwise loss metric fails to take account of the relative sensitivity or otherwise of the output to particular perturbations. In that case the used loss l should just be the whole network loss, but then is it really just pruning like every other pruning method, targeting the output loss, but with some built in finetuning.

Finally the results also back up these observations: the approach is not particularly better than other methods, at least as far as I can tell. This is not helped by the fact there appears to be heavy inconsistency in the experimental methods applied to different models. Experiments on LeNet and MNIST are pretty irrelevant anyway. Likewise VGG is irrelevant as it is a a heavily redundant model, and should play no part in modern pruning experiments: it is destroyed by a much smaller convnet without a FC layer. The experiments seem more consistent with a paper from 8 years ago than a modern setting. Otherwise it is not clear to me how the experiments were conducted. For example, did other methods get finetuning time (which yours effectively has built in)? Basically this approach has to retrain every layer every time (likely to overcome the accumulated error of the approximation at early layers. What loss functions were used? How were thresholds chosen? Were they different at each layer? How were the competitor methods set up? Were you fair to them in the compute involved (you hint that you were not!) I am afraid I do not find the experiments convincing, and the actually implemented method is hidden away. I suspect once it is clearly stated it is going to be uncannily similar to other methods.

Finally every pruning paper should include a comparison against a directly-trained model of modern form and matched reduced size as a baseline benchmark, so demonstrate whether there is actually any benefit from pruning at all by any method.

Altogether then I have the following major concerns: the paper is unclear on the critical element of the MI estimator actually used. The paper is unclear about why layerwise MI maintenance rather than target MI maintenance is at all important. The paper does not make clear what its firm contribution actually is. There are too many cryptic elements in the paper that hide the actual method uses, and there seems to be a divergence between the paper description and the actual implemented algorithm as far as one can tell. The precise algorithm, and specific implementational choices are still not entirely clear at the end of the paper.  There are spurious theorems. There are inconsistently applied experiments, lacking clear information on the experimental paradigm or benchmark, with many experiments in archaic and uninformative settings.

If I am missing something about all these I would love to know. I will certainly revisit this after author feedback.

**Questions:**

Please could you comment on the other aspect. Maintaining information is necessary not sufficient. It is not enough to maintain information; one must maintain representation. Permuting all the nodes in the layer without permuting parameters retains information, but completely destroys network functionality. You cannot escape from commenting on this aspect.

Please could you carefully define your mutual information estimator, and its properties. What would it calculate for the mutual information between X and Y for X~N(0,Y) Y~chi^2(1)? Where is your loss and your f coming from? What is \theta and f_\theta? And how does this relate to what you do downstream which also seems divorced from this.

Why is your actual method better, and realistically different than just about every other layerwise pruning method: track some simple (e.g. squared error) loss, and only ditch stuff if it doesn't make things much worse. From my reading that seems to be what you do?

Please explain your exact experimental paradigm for ResNets on CIFAR10 and TinyImagenet. Why are the experiments not systematic? What are the implementational details of your methods and others? Why are you not using a standardised benchmark for this?

---

> ### Author Response · Authors · 2025-11-28
> **Response**
>
> We thank the Reviewer for the detailed and very useful comments. In the following we address each concern raised in the review.
>
> ### Comment 1
> **Reviewer:**
> The paper references preserving mutual information many times, but it is unclear what MI is being preserved...
>
> **Response:**
> Our goal is to preserve **joint mutual information between the activations of adjacent layers before and after pruning**, i.e., $I(X_{l}; X'_{l+1})$. In other words, the MI between masked adjacent layers after pruning should be equal to that between unmasked adjacent layers before pruning. This quantifies how much of the downstream layer’s representational content is retained after pruning the upstream layer. We use a reconstruction-based proxy consistent with prior work. We have now defined the functions, expectations, and loss formally when they are introduced for the first time.
>
> ---
>
> ### Comment 2
> **Reviewer:**
> Only $I(X_L; Y)$ matters for prediction, so why not prune using only MI with the output?
>
> **Response:**
> Directly estimating MI between each intermediate layer and the output is needlessly difficult particularly for early layers when we already have access to the function that relates layers. By pruning **from outputs backward to inputs**, each earlier layer is constrained to retain only information needed by the already-reduced layer after it. This backward approach implicitly preserves output-relevant information at every layer while avoiding the curse of dimensionality. This motivation is explained in Section 5.3.
>
> ---
>
> ### Comment 3
> **Reviewer:**
> Why is pruning from outputs to inputs preferred?
>
> **Response:**
> We follow Luo et al. (2017) and we have now stated this clearly in Section 5.3: we prune from **outputs → inputs** ensures that irrelevant entropy is not propagated backward.
>
> ---
>
> ### Comment 4
> **Reviewer:**
> Maintaining information is not sufficient.
>
> **Response:**
> We agree with the Reviewer on this point. MI preservation is **necessary but not sufficient**. We have added a statement acknowledging this.
>
> ---
>
> ### Comment 5
> **Reviewer:**
> The theorem(s) seem unnecessary or unhelpful.
>
> **Response:**
> We have removed **Theorem 1** as it restates a classical DPI fact and is not essential.
> We  have retained **Theorem 2**, as it highlights that the MI between data and the pruning mask is fully determined by activations and not by whether pruning is applied. We believe this observation is of technical interest and have clarified the explanation to clarify its role.
>
> ---
>
> ### Comment 6
> **Reviewer:**
> The paper should relate more clearly to prior information-theoretic pruning and feature-selection literature (Dai 2018; Isik; Battiti 1994, etc.).
>
> **Response:**
> We agree with the point raised by the Reviewer. We have expanded the related work section to situate our method directly within this body of work, highlighting how our backward MI-preservation strategy and layer-wise reconstruction differ from and extend these approaches.
>
> ---
>
> ### Comment 7
> **Reviewer:**
> The requirement $I(X_{\ell-1}; X_l) = I(X_{l-1 \setminus i}; X_l)$ is extremely strong.
>
> **Response:**
> We have clarified that this equality is **not assumed operationally**. In practice, we use MI *thresholds* and rely on the empirical observation that **removing individual units often does not change MI significantly**, especially in redundant layers. This theoretical equality is not necessary for the algorithm to operate.
>
> ---
>
> ### Comment 8
> **Reviewer:**
> The mutual information approximation in the paper is cryptic...
>
> **Response:**
> The Reviewer is correct in that the estimator is indeed a proxy for MI. We would like the reviewer to refer to the following paper for more details:
>
> Covert et al. Understanding Global Feature Contributions with Additive Importance Measures. In *NeurIPS'20*, 2020.
> ---
>
> ### Comment 9
> **Reviewer:**
> The expectation underlying the MI estimator cannot be computed...
>
> **Response:**
> We have clarified that the estimator is computed as an **empirical expectation over minibatches**, following standard practice.
>
> ---
>
> ### Comment 10
> **Reviewer:**
> The claim “If the target is discrete and cross entropy is used...
>
> **Response:**
> We agree this statement lacks precision. We have removed and replaced it with a more accurate description.
>
>
> ---
>
> ### Comment 11
> **Reviewer:**
> The novelty is unclear.
>
> **Response:**
> In the paper we highlight that:
> 1. **Backward MI-preserving pruning** ensuring output-relevant information at every layer;
> 2. **Information–mask independence argument (Theorem 2)**;
> 3. The **explicit grounding in MI ordering and TERC**;
> 4. The **use of MI-driven pruning masks** that generalise to feature selection (Appendix G).
>
>
>
> ---
>
> ### Comment 12
> **Reviewer:**
> The implemented method appears to involve retraining each layer, which may be expensive or similar to other methods.
>
> **Response:**
> We have discussed this in Appendix G.
>
> ---

---

### Official Review · Reviewer_4pjw · 2025-11-06

**Soundness:** 3
**Presentation:** 3
**Contribution:** 3
**Rating:** 6
**Confidence:** 4

**Summary:**

This paper presents a pruning methods called Mutual Information Preserving Pruning (MIPP) based on the idea of pruning whilst maintaining the pairwise mutual information between layers, starting from mutual information between the input and the first layer (algorithm 1). For each pair of layers, the pruning is done neuron-wise by zeroing-out one neuron which leave the mutual information with the next layer unchanged (algorithm 2).

Some general, sanity-checking, information-theoretic results are shown to support this method (Theorem 1,2). The actual computation of mutual information is approximated using a method from Covert et al. The authors then apply the methods to a variety of models trained on CIFAR10/100/TinyImageNet., considering both pruning after initialisation and pruning after training scenarios.

**Strengths:**

- The method is principled and grounded in a solid conceptual framework (information theory).
- Whilst mutual information is a nice concept it is not a computationally convenient one. The authors have nevertheless found a way to approximate this computation and make their pruning algorithm scale (at least seeminly, see question below).
- From the empirical evaluation it is clear that the proposed method works well, at least on the kind of task and architecture being considered.
- The proposed method may be helpful beyond pruning, particularly in providing a functional interpretation of neural networks in terms of information flow.
- The writing is overall pretty good.

**Weaknesses:**

Some technical aspects should be tightened/cleaned up a bit. Specifically:
- 172: I would expect a Hadamard product $\odot$ here. Also, only a very particular kind of pruning is considered here: the mask is not an $m\times n$ matrix but an $n\times 1$ vector, which is equivalent to having an $m\times n$ mask where entire columns are set to zero, a very special kind of $m\times n$ mask.
- 207: what exactly do you mean by “by gradient ascent”?
- 207: what is $\mathcal{F}$? Is this related to the previous question?
- Theorem 1: the proof is not very convincing, or not very well-explained (partly because $\mathcal{F}$ is not defined anywhere). The variational (Donsker–Varadhan) representation of mutual information is not stated very well (the expectations are not taken w.r.t. the same measures, $f$ takes two inputs $X,Y$). Are the if and only if directions proved simultaneously? I.e. is there a chain of equivalences in this proof?
- Theorem 2: the statement is not quite correct, I think the inequality should be an equality $I(\mathcal{D},\mathcal{M})=H(\mathcal{D})$ else the maximum is not reached. The proof is also unconvincing, or badly presented. Are both the if and only if parts presented at the same time? Where is the layer-wise assumption used in the if direction?
- 270-276: this paragraph is very sloppily presented. I managed to make some sense of it with the help of Covert et al. In particular $f_\emptyset(\emptyset)$ should be $f_{\emptyset}(X_{\emptyset})$ which is just $\mathbb{E}[f(X)]$ and $f_X$ is just the model $f$ (this must be stated or line 288 makes no sense). I cannot find any proof reference for the $\approx$ statement on line 273 in Covert et al.
- 286: h is introduced 10 lines below!
- Fig 5-6 are pretty much unreadable. Perhaps choose only one of the two?
- 402: “each layer in a neural network is an injective function of its predecessor” please explain how this makes sense, because on the face of it this cannot be true.

**Questions:**

Can you give me some sense of how computationally expensive MIPP is. The fact that the network gets locally retrained as many times as there are neurons in the network sounds quite costly, but then you have results over reasonably large networks. Is it more expensive in the PaI than in the PaT scenario?

---

> ### Author Response · Authors · 2025-11-28
> **Response**
>
> ### Comment 1
> **Reviewer:**
> Line 207: what exactly is meant by “gradient ascent”?
>
> **Response:**
> We simply mean applying gradient descent to the negative of the loss function. We will rephrase the statement to make this clearer and avoid ambiguity.
>
>
> ---
>
> ### Comment 2
>
> **Reviewer:**
> Line 207: what is $\mathcal{F}$? Is this related to the previous question?
>
> **Response:**
> $\mathcal{F}$ denotes the set of all candidate functions under consideration. We will define this explicitly when it first appears.
>
> ---
>
> ### Comment 3
> **Reviewer:**
> Theorem 1: the proof is unconvincing and poorly explained, partly because $\mathcal{F}$ is undefined. The Donsker–Varadhan representation is not stated correctly, and the expectations appear taken with respect to different measures. Are the “if” and “only if” parts proved simultaneously?
>
> **Response:**
> We appreciate the Reviewer’s detailed feedback. We agree that the statement was problematic. We have removed this theorem and its proof from the manuscript, as it does not contribute substantively to the method.
>
>
> ### Comment 4
> **Reviewer:**
> Theorem 2: the inequality should be an equality; otherwise the maximum is not attained. The proof is unconvincing and unclear. Are both the forward and reverse directions being shown at once? Where is the layer-wise assumption used?
>
> **Response:**
>
> Could you clarify exactly which inequality and specific assumption you are referring to? The inequality in the statement of Theorem 2 is intentional and correct as written. The purpose of this theorem is to demonstrate that, rather than merely preserving the information contained in the low-entropy masks $\mathcal{M}$, our method also preserves the information present in the high-entropy data $\mathcal{D}$. This result provides a counterpoint to Kumar et al., who argue that the information in the data cannot be accessed prior to training [1].
>
> [1] Kumar, A., et al. "No Free Prune: Information-Theoretic Barriers to Pruning at Initialization." *ICML*, 2024.
>
> ### Comment 5
> **Reviewer:**
> Lines 270–276: the paragraph is very sloppily presented. For example, $\hat{f}$ should be $\hat{f}_X$, which is just the model $f_\theta$. The quantity on line 273 does not seem to appear in Covert et al., and the statement seems incorrect.
>
> **Response:**
> We agree that this section requires clearer presentation. Line 273 is indeed incorrect and will be removed. We will rewrite the entire paragraph to use consistent notation, properly introduce $f_\theta$ when first needed, and ensure that all references to Covert et al. are accurate.
>
>
> ---
>
> ### Comment 6
> **Reviewer:**
> \(h\) is introduced 10 lines below its use.
>
> **Response:**
> We will correct this by defining \(h\) immediately upon first use.
>
> ---
>
> ### Comment 7
> **Reviewer:**
> Line 402: “each layer in a neural network is an injective function of its predecessor” — this appears false. Please explain.
>
> **Response:**
> We acknowledge that real neural networks are not injective in practice. The statement refers only to an idealised theoretical construction used to illustrate certain information-theoretic properties. We will revise the text to clarify that this is a conceptual device rather than an empirical claim.
> ### Comment 8
> **Reviewer:**
> What is the computational cost of MIPP? Retraining the network as many times as there are neurons seems expensive. Is PaI more costly than PaT?
>
> **Response:**
> We include computational-cost measurements in Appendix G.

---

### Meta-Review · Area_Chair_3afM · 2026-01-07

**Summary:**

All reviewers raised concerns about clarity, implementation specifics, overclaiming of theoretical results, and lack of novelty relative to prior pruning work. Despite some empirical results, and the experimental setup lacks rigor and modern benchmarks. The rebuttal addressed presentation issues but did not resolve concerns about core contributions, MI estimation, or reproducibility.

**Reviewer Concerns:**

Major concerns remain unresolved:

The mutual information estimator is poorly defined.

The practical method resembles conventional layer-wise loss-based pruning.

The conceptual benefit of preserving inter-layer MI (rather than output-relevant MI) remains unmotivated.

Experiments lack standardization, rigorous baselines, or comparisons on modern tasks/models (e.g., ViT, ImageNet-1K).

Several empirical claims (e.g., retrainability) are contradicted by the paper’s own figures.

**Reviewer Scores:**

4pjw (6 → 6): Appreciated conceptual framing but raised technical and theoretical issues. Rebuttal helped, but likely still views the work as borderline.

Y8wy (2 → 2): Major doubts about the core novelty, MI estimation, and practical distinctiveness of the method remain unaddressed.

huq5 (2 → 2): Strongly critical of experimental inconsistencies, retrainability claims, and evaluation protocol. Rebuttal acknowledged but not persuasive.

z8gy (4 → 4): Recognized potential value, but was unconvinced by outdated baselines and weak generalization to modern models. Score likely unchanged.

---

### Decision · Program_Chairs · 2026-01-26

Reject